# A SAM-key domain required for enzymatic activity of the Fun30 nucleosome remodeler

Leonhard A Karl[1], Lorenzo Galanti[1,2,3], Susanne CS Bantele[1], Felix Metzner[4], Barbara Šafarić[5], Lional Rajappa[5], Benjamin Foster[6], Vanessa Borges Pires[3], Priyanka Bansal[7], Erika Chacin[7], Jerôme Basquin[8], Karl E Duderstadt[5,9], Christoph F Kurat[7], Till Bartke[6], Karl-Peter Hopfner[4], Boris Pfander[1,2,3]

Fun30 is the prototype of the Fun30-SMARCAD1-ETL subfamily of nucleosome remodelers involved in DNA repair and gene silencing. These proteins appear to act as single-subunit nucleosome remodelers, but their molecular mechanisms are, at this point, poorly understood. Using multiple sequence alignment and structure prediction, we identify an evolutionarily conserved domain that is modeled to contain a SAM-like fold with one long, protruding helix, which we term SAM-key. Deletion of the SAM-key within budding yeast Fun30 leads to a defect in DNA repair and gene silencing similar to that of the *fun30Δ* mutant. In vitro, Fun30 protein lacking the SAM-key is able to bind nucleosomes but is deficient in DNA-stimulated ATPase activity and nucleosome sliding and eviction. A structural model based on AlphaFold2 prediction and verified by crosslinking-MS indicates an interaction of the long SAM-key helix with protrusion I, a subdomain located between the two ATPase lobes that is critical for control of enzymatic activity. Mutation of the interaction interface phenocopies the domain deletion with a lack of DNA-stimulated ATPase activation and a nucleosome-remodeling defect, thereby confirming a role of the SAM-key helix in regulating ATPase activity. Our data thereby demonstrate a central role of the SAM-key domain in mediating the activation of Fun30 catalytic activity, thus highlighting the importance of allosteric activation for this class of enzymes.

## Introduction

Nucleosome remodelers are ATP-driven molecular machines of the superfamily 2 (SF2) of DNA translocases (Flaus et al, 2006) that govern the locations of nucleosomes on DNA to dynamically shape chromatin (Becker & Workman, 2013; Clapier et al, 2017). Therefore, remodelers catalyze the sliding, eviction, and positioning of nucleosomes and also edit nucleosomes by catalyzing histone exchange. Remodelers use energy from ATP hydrolysis via a conserved two-lobed Swi2/Snf2-type ATPase domain to break contacts between DNA and histones. To facilitate nucleosome remodeling, additional interactions with DNA and histone proteins are necessary (Clapier et al, 2017; Dao & Pham, 2022). Furthermore, additional elements within remodelers are required for their recruitment and to regulate their activity in specific chromatin regions. Notably, several remodelers form megadalton multi-protein complexes, whereas others appear to act as single-subunit enzymes (Clapier et al, 2017). Studying single-subunit remodelers does not only have the potential to reveal the critical elements of the particular enzyme, but also to conceptualize minimal elements required for remodeler function.

The Fun30-SMARCAD1-ETL subfamily of remodelers is made up of single-subunit remodelers with broad cellular functions (Bantele & Pfander, 2019; Karl et al, 2022). Two major functions appear to be evolutionarily conserved from yeast (budding yeast Fun30, fission yeast Fft3) to human (SMARCAD1): first, both budding yeast Fun30 and human SMARCAD1 function in the DNA damage response, where they have been shown to promote DNA end resection of DNA double-strand breaks (DSBs) and repair by homologous recombination (Chen et al, 2012; Costelloe et al, 2012; Eapen et al, 2012; Bantele et al, 2017); second, Fun30, SMARCAD1, and fission yeast Fft3 have a role in the maintenance of silent chromatin. Upon deletion of *FUN30*, transcriptional silencing is lost from telomeric and silent mating type loci (Neves-Costa et al, 2009; Durand-Dubief et al, 2012). Similarly, in the absence of Fft3 in fission yeast, transcriptional silencing and heterochromatin structure are lost from centromeres and sub-telomeres (Steglich et al, 2015). Lastly, human SMARCAD1 is

[1]DNA Replication and Genome Integrity, Max Planck Institute of Biochemistry, Martinsried, Germany   [2]Genome Maintenance Mechanisms in Health and Disease, Institute of Aerospace Medicine, German Aerospace Center (DLR), Cologne, Germany   [3]Genome Maintenance Mechanisms in Health and Disease, Institute of Genome Stability in Ageing and Disease, CECAD Research Center, University of Cologne, Cologne, Germany   [4]Gene Center, Department of Biochemistry, Ludwig-Maximilians-Universität, Munich, Germany   [5]Structure and Dynamics of Molecular Machines, Max Planck Institute of Biochemistry, Martinsried, Germany   [6]Institute of Functional Epigenetics (IFE), Helmholtz Zentrum München, Neuherberg, Germany   [7]Biomedical Center Munich (BMC), Division of Molecular Biology, Faculty of Medicine, Ludwig-Maximilians-Universität in Munich, Martinsried, Germany   [8]Crystallization Facility, Max Planck Institute of Biochemistry, Martinsried, Germany   [9]Physik Department, Technische Universität München, Munich, Germany

Correspondence: bpfander@biochem.mpg.de; boris.pfander@dlr.de
Benjamin Foster's present address is Department of Biochemistry, University of Oxford, Oxford, UK

required for the maintenance of pericentric heterochromatin (Rowbotham et al, 2011). Overall, these roles appear to be linked to a function in the maintenance of chromatin during DNA replication (Rowbotham et al, 2011; Taneja et al, 2017).

The exact substrate of Fun30-SMARCAD1-ETL remodelers and their enzymatic mechanisms remain uncertain. In vitro, Fun30 can slide canonical nucleosomes (Awad et al, 2010), but it can also evict and exchange histone H2A–H2B dimers (Awad et al, 2010). Also, SMARCAD1 shows eviction activity in vitro, but is also able to deposit histone octamers on DNA (Markert et al, 2021). In addition, there is evidence that these remodelers may act on nucleosomes that associate with multivalent nucleosome binders, such as human 53BP1 and budding yeast Rad9 (Bantele & Pfander, 2019; Lo et al, 2021; Karl et al, 2022). Notably, the catalytic activity of Fun30 appears to be stimulated in vitro by single-stranded (ss) and double-stranded (ds) DNA and nucleosomes, but it is unclear whether this reflects activation by substrate binding, as has been observed for other remodelers (Zhou et al, 2016; Adkins et al, 2017; Clapier et al, 2017), or allosteric activation.

Several motifs and binding surfaces have been characterized in Fun30-SMARCAD1-ETL remodelers. These elements include (i) N-terminal CDK phosphorylation sites that mediate binding to Dpb11/TOPBP1 and the 9-1-1 complex and are therefore crucial for recruitment and activation of the remodeler to sites of DSBs (Chen et al, 2016; Bantele et al, 2017, see Fig S1A); (ii) an N-terminal PCNA-binding site in SMARCAD1 that is required for recruitment to sites of DNA replication (Rowbotham et al, 2011; Lo et al, 2021); (iii) a conserved CUE domain (tandem in SMARCAD1, single in Fun30) that in SMARCAD1 was shown to interact with ubiquitinated H2A (Densham et al, 2016) and KAP1 (Rowbotham et al, 2011; Ding et al, 2018; Lim et al, 2019), whereas its binding partner in the yeast protein is still unknown (Awad et al, 2010); and (iv) C-terminal phosphorylation and ubiquitination sites in SMARCAD1 that are required for SMARCAD1 function at DSBs, but are seemingly not conserved in lower eukaryotes (Chakraborty et al, 2018). Notably, a commonality of these elements is that they lead to binding and recruitment to specific DNA replication and repair proteins or to specific chromatin regions. Although recruitment is certainly an important mechanism to regulate these remodelers, it appears to be separate from the actual mechanism of catalysis (see below). Therefore, with the exception of the two-lobed ATPase domain, which is essential for all functions of Fun30-SMARCAD1-ETL remodelers, we do not know of any additional motifs or domains that might give insight into the molecular mechanisms of these enzymes. In addition, although a low-resolution cryo-EM map has been obtained for nucleosome-bound SMARCAD1 that may suggest an unconventional mode of binding to the nucleosome dyad (Markert et al, 2021), these data did not have sufficient resolution to identify key structural elements of Fun30-SMARCAD1-ETL remodelers.

To identify elements within budding yeast Fun30 that are critical for its catalytic function, we took a dual-screening approach. Testing truncations of Fun30 in a functional assay for DNA repair pointed towards a critical region located between CUE and ATPase domains. This region is evolutionary conserved and, according to structure predictions, forms a domain that we call SAM-key. Deletion of the SAM-key abolishes Fun30 functions in DNA damage

response and gene silencing in vivo, and DNA-stimulated ATP hydrolysis and nucleosome remodeling in vitro. We verified an AlphaFold2 model using crosslinking-MS (XL-MS), which showed that the SAM-key interacted with protrusion I of the Fun30 ATPase. Structural alignment of the Fun30 model with different nucleosome-bound remodeler structures revealed structural similarities of the SAM-key to the post-HSA helix in Ino80, which is involved in protrusion I interaction as well. Mutation of the SAM-key–protrusion I interaction surface in Fun30 phenocopied the SAM-key domain deletion, suggesting that allosteric activation of the remodeler by the SAM-key domain is required for Fun30 enzymatic activity. As such, the SAM-key may fulfill similar functions as related but more complex modules in other nucleosome-remodeling complexes.

## Results

### Identification of the SAM-key domain in Fun30-SMARCAD1-ETL remodelers

To define modules required for the catalytic activity of the Fun30-SMARCAD1-ETL remodelers, we undertook a dual approach based on functional in vivo assays and homology-based structural modeling. First, we tested whether truncations of budding yeast *FUN30* would support its function in DNA repair in yeast (Fig 1A and B). Apart from the ATPase domain, previous work has identified several crucial elements in Fun30-SMARCAD1-ETL remodelers, all of which, however, appear to act at the stage of chromatin recruitment (Awad et al, 2010; Rowbotham et al, 2011; Chen et al, 2016; Densham et al, 2016; Bantele et al, 2017; Ding et al, 2018; Lim et al, 2019; Lo et al, 2021, see Introduction section). To target our screen to elements required for catalytic activity rather than recruitment, we screened a *DDC1–FUN30* fusion construct, which we have previously shown to bypass endogenous recruitment elements targeting Fun30 to sites of DNA damage (Bantele et al, 2017). We also left the C-terminal ATPase domain intact and introduced different truncations to the N-terminal and central regions of the protein. To assay for Fun30's DSB repair function, we tested sensitivity to camptothecin (CPT) (Fig 1A), and resection of a non-repairable DSB induced at MAT by the HO-endonuclease using RPA-ChIP–qPCR (Fig 1B). We found that in the context of the *DDC1–FUN30* fusion, the N-terminus of Fun30 (until aa 120, including CDK phosphorylation sites and CUE domain) was dispensable for DSB repair (Fig 1A and B), even though it is otherwise required for Fun30 recruitment and function (Bantele et al, 2017). In contrast, the central part of Fun30 was required for its repair function (Fig 1A and B), and even short truncations such as Δ338–389 abolished Fun30's DSB repair function (Fig S1A), suggesting this region may be required for Fun30 activity.

In parallel, we conducted multiple sequence alignments of Fun30-SMARCAD1-ETL remodelers from different eukaryotes and found a region of high-sequence conservation in the center of the protein from aa 279–387, which was previously uncharacterized (Figs 1C and S1B). Structure predictions using AlphaFold2 (Jumper et al, 2021) indicated with high confidence that this region would fold in a domain that contains a sterile alpha motif (SAM)-like fold,

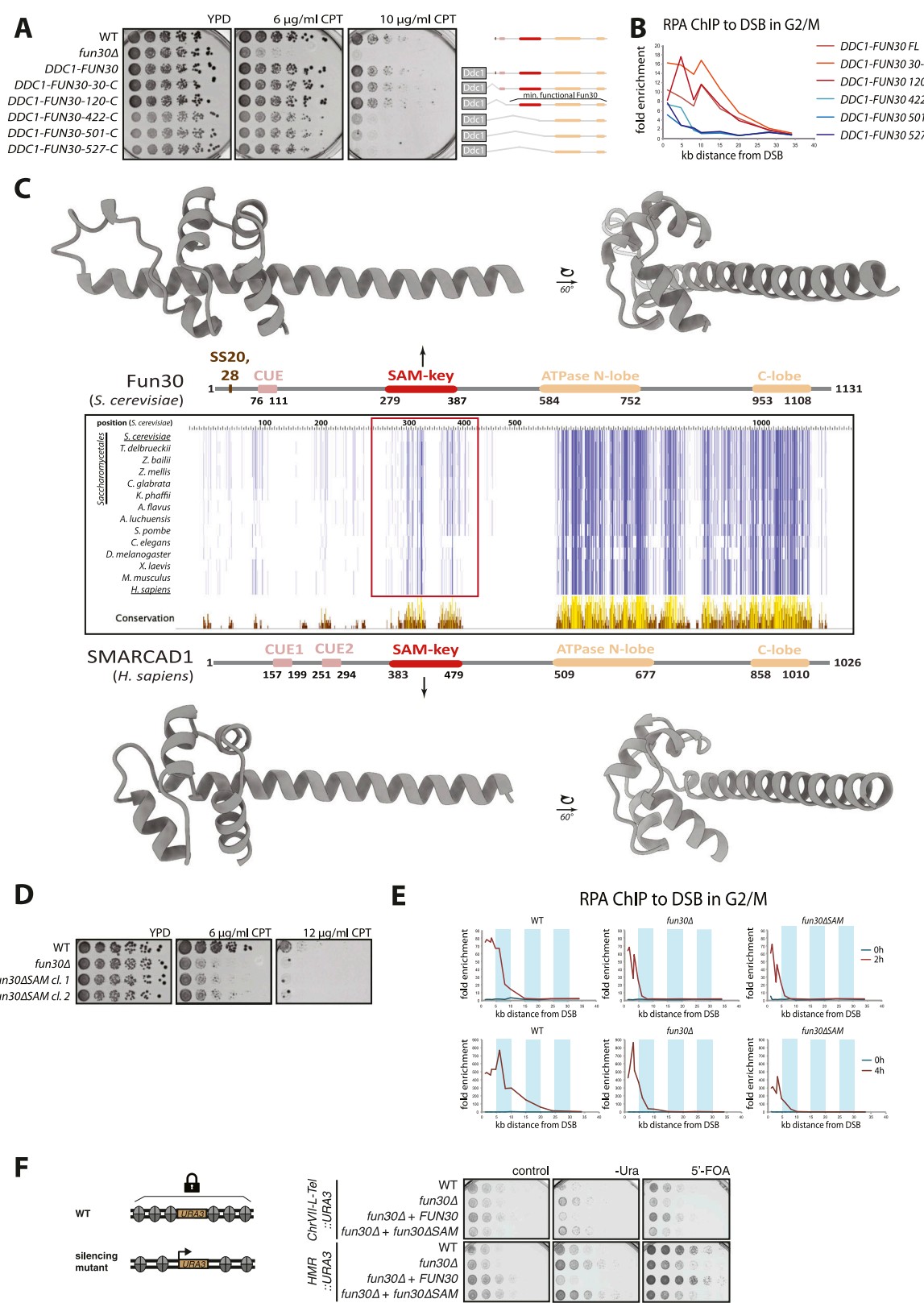

**Figure 1. SAM-key—a conserved domain required for function of nucleosome remodeler Fun30.**
**(A, B)** A series of N-terminal truncations in Fun30 reveals a previously uncharacterized, central (aa 120–422) region important for DNA damage repair function.
**(A)** Sensitivity to different dosages of camptothecin of budding yeast cells expressing N-terminal truncations of Fun30 in a *fun30Δ* background, but in the context of a covalent *DDC1–FUN30* fusion that forces the recruitment to DNA damage sites. Truncations included or excluded respectively: the CDK-phosphorylation sites SS20, 28 (dark

but with its C-terminus extended by a long alpha helix (confidence per residue estimation, pLDDT[Fun30] = 81.97; Fig 1C). Based on its fold, we called this domain SAM-key. Sequence conservation was highlighted by the fact that human SMARCAD1 is also predicted to form a SAM-key by AlphaFold2 (pLDDT[SMARCAD1] = 94.53; Fig 1C).

We deleted the SAM-key (Δ275–426) in *FUN30* expressed from the endogenous promoter (*fun30ΔSAM*). The truncation does not affect Fun30 expression (Fig S1F). However, deletion of the SAM-key phenocopied the deletion of the *FUN30* gene in its DSB repair function as demonstrated by pronounced sensitivity to CPT (Fig 1D), and reduced spreading of resection at a non-repairable DSB (Fig 1E). These defects were comparable with what can be observed with the *fun30Δ* strain (Fig 1D and E, see also Chen et al, 2012; Bantele et al, 2017). A similar defect was also observed in the background of the *DDC1–FUN30* fusion constructs, indicating that the SAM-key is required for DSB repair even in the context of the fusion protein that is forced to localize to sites of DNA damage (Fig S1D and E).

Given that the SAM-key was required for Fun30's function in response to DNA damage, we also tested involvement in Fun30's second major function in yeast—gene silencing (Neves-Costa et al, 2009). To this end, we used *URA3*-based genetic silencing reporters (Singer et al, 1998; Meijsing & Ehrenhofer-Murray, 2001; Neves-Costa et al, 2009) integrated at two distinct silenced loci: the telomere on the left arm of chromosome 7 and the silent mating-type locus HMR (Fig 1F). Upon loss of silencing of these loci, the *URA3* reporter gene will be expressed resulting in enhanced growth on synthetic medium lacking uracil (-Ura), but reduced growth on medium containing 5'-FOA. Interestingly, the *fun30ΔSAM* strains showed silencing defects at both telomeric and silent mating-type loci, similar to what was observed in the *fun30Δ* or catalytically inactive *fun30-K603R* strains (Figs 1F and S1C). Overall, we therefore conclude that the SAM-key domain is conserved in Fun30-SMARCAD1-ETL remodelers and required for major Fun30 functions.

### The SAM-key is not required for Fun30 binding to DNA and nucleosomes

To biochemically characterize how the SAM-key may affect Fun30 function, we developed strategies to purify Fun30, Fun30ΔSAM, and catalytically inactive Fun30-K603R (Walker A mutation) after heterologous expression in bacteria and overexpression in yeast

(Fig S2A and B). Purification after bacterial expression involved two steps of affinity purification followed by cleavage of purification tags and gel filtration of untagged proteins (Fig S2A and B). Limited proteolysis with five different proteases revealed similar cleavage patterns of Fun30 and Fun30ΔSAM, suggesting that overall folding of both proteins was comparable (Fig S2C). In support of this, we obtained very similar yields and concentrations of both proteins (Fig S2A), and nano differential scanning fluorimetry revealed similar melting points (Fig S2D).

We performed gel-shift analysis to test whether the SAM-key may influence the binding of Fun30 to DNA or nucleosomes. First, we tested Fun30 binding to double-stranded (ds) DNA (100W0; 247 bp). The disappearance of the free DNA suggested that Fun30 bound to DNA at high nanomolar concentrations (Figs 2A and S2E). Although the Fun30–DNA complex could not be resolved as a discrete band, it was reversible by the addition of competitor DNA, suggesting it was no aggregation (Fig S2F). Notably, this binding was not influenced by the deletion of the SAM-key (Figs 2A and S2E). Next, we tested binding to mononucleosomes that were end-positioned on the same dsDNA with a 100-bp overhang (100W0) and labelled on histone H2A with the fluorophore Dylight 550 (Safaric et al, 2022). A large proportion of these nucleosomes were bound by Fun30 in the nanomolar concentration range (Figs 2B and S2G). Nucleosome binding was reversible (Fig S2H), but, importantly, independent of the SAM-key domain (Fig 2B). SAM-key-independent binding of Fun30 to nucleosomes was also confirmed by in vitro coIPs of nucleosomes using tagged versions of Fun30 and Fun30ΔSAM (Fig 2C). In summary, we therefore conclude that Fun30 binding to its nucleosome substrate is largely intact in the absence of the SAM-key.

### The SAM-key is required for nucleosome remodeling by Fun30

Given that deletion of the SAM-key resulted in a loss-of-function phenotype in vivo, we tested for Fun30 catalytic activity in vitro. Previous work had shown that purified Fun30 is able to slide end-positioned nucleosomes on dsDNA to a more central position (Byeon et al, 2013) in an ATP-dependent reaction. Similarly, we observed also in our hands using end-positioned nucleosomes (100W0) that Fun30 slid nucleosomes and positioned them more centrally in a reaction that required ATP hydrolysis (Figs 3A and B and S3D, seen for both H4- and H2A-labelled nucleosomes).

---

brown), the CUE-domain (rosy brown), and differently sized fragments of the N-terminal part of the protein. All constructs contained the C-terminal part with the conserved SNF2-type two-lobed ATPase domain (beige). The truncation constructs starting at residue 422 (422-C) or even further towards the C-terminus show increased sensitivity, similar to *fun30Δ*. **(B)** The pGal:HO system was used to induce a single DSB in G2/M phase at the MAT locus in yeast strains carrying truncated *DDC1–FUN30* fusion constructs as in (A). Spreading of resection as measured by RPA ChIP qPCR to the DSB shows over-resection phenotype for the *DDC1–FUN30* fusion and minimal truncations (shades of red), whereas resection is defective in truncations starting at residue 422 (422-C) and thereafter (shades of blue). **(C)** Multiple sequence alignments of Fun30/SMARCAD1/ETL orthologues reveal the SAM-key domain. The central panel shows alignment of the full-protein sequences with conserved residues indicated by dark blue color using ClustalWS. 2D representations of domain architecture of budding yeast Fun30 (above) and human SMARCAD1 proteins (below) with CDK-phosphorylation sites SS20, 28 (dark brown), the CUE-domain(s) (rosy brown), the SAM-key (red), and the conserved SNF2-type two-lobed ATPase domain (beige). AlphaFold2 predictions of the SAM-key domain are shown as 3D models on top (yeast) and bottom (human). **(D, E, F)** Truncation of the SAM-key abolishes Fun30 function in DSB repair and gene silencing. **(D)** Sensitivity to different dosages of camptothecin of WT, *fun30Δ*, and *fun30ΔSAM* (Δ275–436) budding yeast cells in the growth assay. n = 3 biological replicates. **(E)** Spreading of resection was measured as in (B) and is defective in *fun30ΔSAM* (Δ275–436) cells. RPA ChIP qPCR to a single, induced DSB in the G2/M phase shows reduction of resection spreading and slower kinetics for *fun30Δ* and *fun30ΔSAM* strains compared with WT. Upper panels show 0 and 2 h timepoints, lower panels show 4 h. **(F)** Gene silencing assay: the *URA3* auxotrophic marker is integrated in a transcriptionally silenced genomic location (telomere [upper], silent mating type [HMR, lower]), upon loss of silencing, *URA3* is expressed allowing growth on -Ura medium, but not on 5'-FOA. A *fun30Δ*-silencing defect is rescued by expression of Fun30 WT, but not Fun30ΔSAM protein. n = 3 biological replicates.

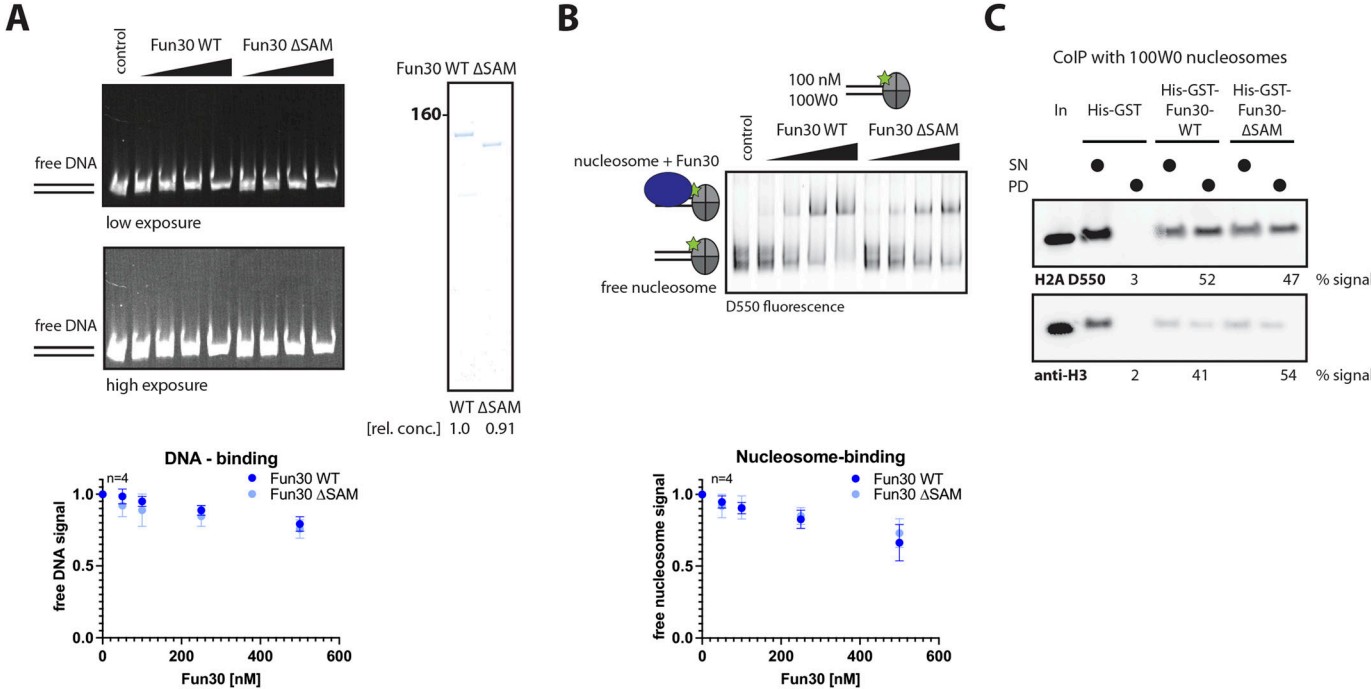

**Figure 2. The SAM-key domain is not required for Fun30 binding to DNA or nucleosomes.**
**(A)** SAM-key is not required for DNA binding. Left: representative gel picture showing binding of purified Fun30 and Fun30ΔSAM in native gels with a 247-bp dsDNA construct, carrying an end-positioned Widom 601 nucleosome-positioning sequence (100W0) stained by ethidium bromide and protein titration (50, 100, 250, and 500 nM Fun30). Low exposure (upper) and high exposure (lower) are shown to visualize shifted species. Coomassie gel (right) shows equivalent amounts of input protein as quantified from band intensity (same for (B), DNA and nucleosome-binding experiments were performed side-by-side). Bottom: quantification of free DNA in presence of Fun30 WT (dark blue) and Fun30ΔSAM (light blue) normalized to the control lane (without remodeler). n = 4 replicates, filled circles indicate the mean, error-bars depict SD. Individual datapoints of replicates are shown in Fig S2E. **(B, C)** SAM-key is not required for nucleosome binding. **(B)** Gel-shift assay as in (A), but with yeast nucleosome end-positioned on 100W0 DNA. Histone H2A was labeled with Dylight550 (Thermo Fisher Scientific) at cysteine 46 (H2A 46-C-D550). Top: representative gel picture, bottom: quantification as in (A), but this time, for free nucleosome signal. n = 4 replicates, filled circles indicate the mean, error bars depict SD. Individual datapoints of replicates are shown in Fig S2G. **(C)** Nucleosome pulldown with His-GST-Fun30 WT or His-GST-Fun30ΔSAM and a tag-only construct (IP for GST) and reconstituted yeast nucleosomes. Western blot for histone H3 and fluorescence imaging of labeled H2A (H2A 46-C-LD550) show that both Fun30 proteins bind comparably to nucleosomes. Percentage numbers below indicate quantification of the signal in the pulldown band relative to total signal.
Source data are available for this figure.

Titrating Fun30 concentration, we observed that Fun30 was able to catalyze the sliding reaction, but neither the catalytic inactive K603R mutant (Walker A) nor the deletion of the SAM-key was able to support this reaction (Fig 3C).

It has been argued that histone dimer or octamer eviction is a key enzymatic activity of Fun30-SMARCAD1-ETL remodelers (Awad et al, 2010; Markert et al, 2021). To measure eviction, we employed the histone chaperone Nap1, which is known to bind H2A–H2B dimers and H3–H4 tetramers (McBryant et al, 2003). When added to remodeling reactions, Nap1 functions as a sink for evicted histone H2A–H2B, and we can therefore follow H2A–H2B eviction with fluorescently labelled H2A (Fig 3D). In this assay, we can detect eviction not only by the loss of the nucleosome signal but also the appearance of a labelled H2A–H2B in complex with Nap1 and free DNA (Figs 3E–G and S3E). Specifically, we observed that eviction is dependent on Fun30 in a concentration- and ATP hydrolysis-dependent manner (Fig 3E). Moreover, H2A–H2B eviction was abolished in Fun30ΔSAM and Fun30-K603R mutant proteins, showing that also in this context, the SAM-key was required for remodeling activity (Fig 3F and G). Whereas Fun30 was shown to have H2A–H2B dimer exchange activity, the occurrence of nucleosome-free DNA suggested that in

the context of our assay, nucleosomes were entirely removed from DNA. Therefore, we measured the eviction of H3–H4 tetramers using labelled H4 (Fig 3F) and also in this case, observed eviction with WT Fun30, but not with Fun30ΔSAM protein.

As a third read-out of Fun30 activity, we measured ATP hydrolysis by Fun30. To this end, we used a NADH-coupled, colorimetric assay to measure ATP hydrolysis rates at steady state (Forné et al, 2012). This assay showed very low ATP hydrolysis by isolated Fun30 ($k_{cat}$ below 0.3 $s^{-1}$), but different constructs of single-stranded and double-stranded DNA stimulated ATP hydrolysis by Fun30 up to $k_{cat}$ of 3 $s^{-1}$ (Fig S3A). Notably, when we compared Fun30ΔSAM with the full-length protein, we found that the SAM-key was required for DNA-stimulated ATP hydrolysis (Figs 3H and I and S3B). As such, we conclude that the SAM-key is either intrinsically required for ATP hydrolysis by the ATPase domain or that it is critical to allosterically activate the ATPase (see below).

We therefore wondered whether the SAM-key could only function in cis as part of the same polypeptide chain or whether addition of the isolated SAM domain could restore catalytic activity of Fun30ΔSAM in trans. Sufficient amounts of soluble SAM-key (aa 275–436) could be expressed and purified from bacteria (Fig S3C).

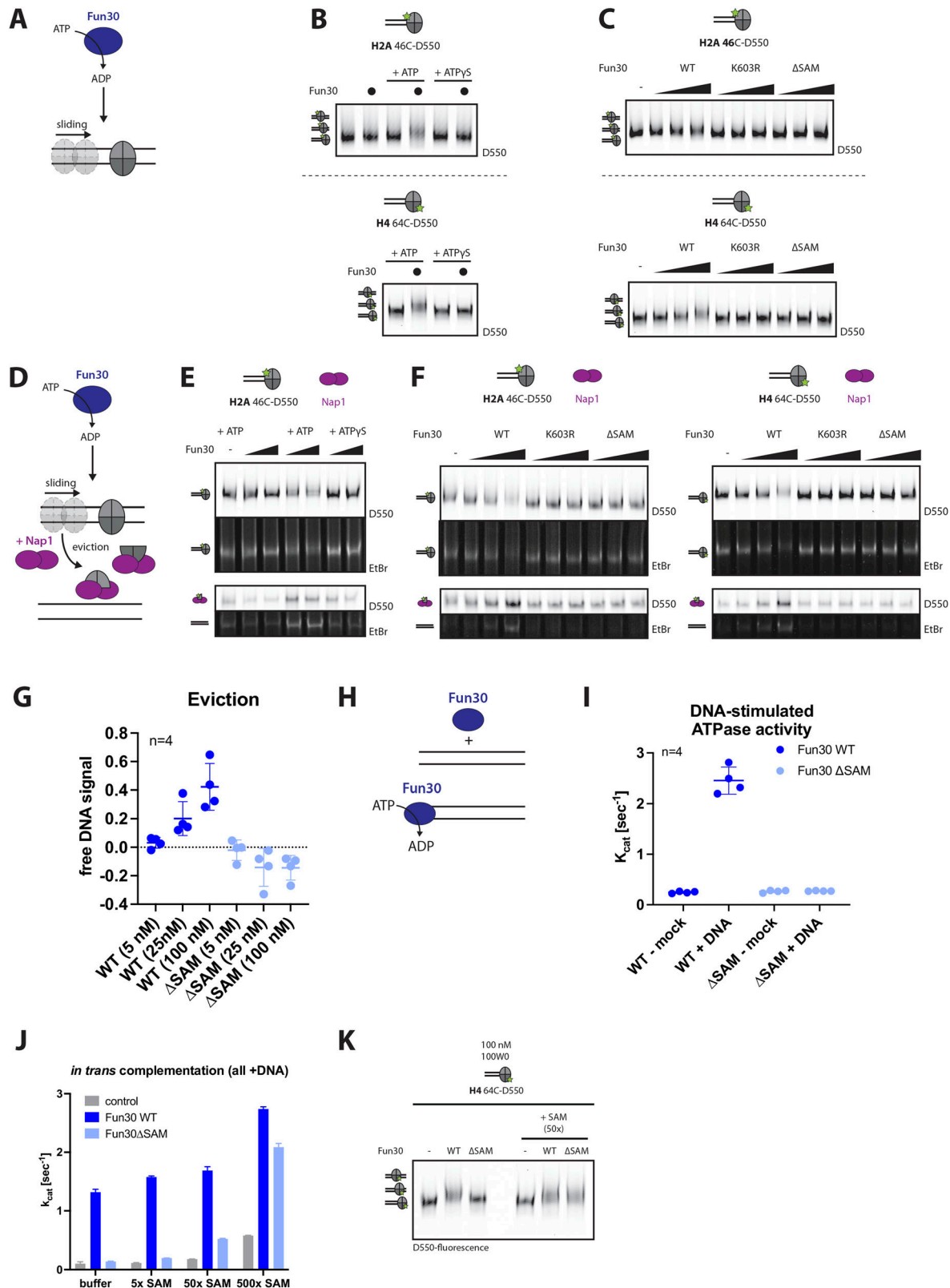

**Figure 3. The SAM-key is required for Fun30 nucleosome remodeling.**
**(A, B, C)** The SAM-key is required for nucleosome sliding. **(A)** Schematic of nucleosome sliding assay: an end-positioned nucleosome is mobilized towards the center of a DNA fragment catalyzed by remodeler in an ATP-dependent fashion. **(B)** Sliding of nucleosomes with labeled H2A (46-C-D550, top) or labeled H4 (64-C-D550, bottom) assembled on a 100W0 fragment (247 bp fragment with end-positioned Widom 601-positioning-sequence) as seen by upshift in gel. Shown is ATP-dependence as addition

Addition of excess SAM-key could rescue the ATPase defect of Fun30ΔSAM (Fig 3J) and also the nucleosome-sliding defect (Fig 3K). We therefore conclude that at sufficiently high concentrations, the isolated SAM-key can bind to Fun30ΔSAM and restore its function.

### Crosslinking-MS confirms a structural model of Fun30 including a protrusion I–SAM-key-binding interface

In the absence of any full Fun30 structure and to identify the mechanism by which the SAM-key affects nucleosome remodeling by Fun30-SMARCAD1-ETL remodelers, we turned to a structural model obtained using AlphaFold2 (Jumper et al, 2021; Varadi et al, 2022). The AlphaFold2 model of Fun30 shows the structure of the two-lobed ATPase domain characteristic for Swi2/Snf2 proteins (Fig 4A and B, beige), including protrusion I (Fig 4A and B, orange). In the model, the N-terminal half of the protein is largely unstructured with the exception of CUE (rosy brown) and SAM-key (red) domains (Fig 4A and B). Notably, the model shows an interaction surface between SAM-key and protrusion I (Fig 4B and C). Specifically, this interaction surface involves the long SAM-key helix and two helices of protrusion I (aa 757–797) and appears to be largely hydrophobic (Fig 4B, red and orange highlighted residues). Notably, AlphaFold-Multimer (Jumper et al, 2021) modelled the same interaction surface when we provided SAM-key and Fun30ΔSAM as separate polypeptide chains, as is the case in the in-trans-complementation scenario (Fig S4A–C).

To verify these in silico models, we conducted crosslinking-MS (XL-MS) using the lysine-selective crosslinker BS3. We tested full-length Fun30 in the absence or presence of ATP (Fig 4C), and using the in-trans-complementation conditions with SAM-key added to Fun30ΔSAM (Fig S4C). All conditions gave a similar number (135–216) and overall pattern of crosslinks (Fig S4C), consistent with the observations that ATP typically induces only small conformational changes in the Swi2/Snf2 domains. We therefore used the XL-MS datasets to verify the AlphaFold2 model predictions. Specifically, we filtered for crosslinks between amino acids that were located in structured parts of the model and tested whether those crosslinks would satisfy a 35-Å distance threshold in the AlphaFold2 model of Fun30 (Fig 4C). We found several crosslinks connecting the CUE domain to other parts of the protein that do not satisfy the distance constraint (colored in grey, Fig 4C), suggesting that the CUE domain might be wrongly positioned in the model and/or that its location within the overall Fun30 structure is flexible (Fig 4C). In contrast, the other crosslinks (40) fulfilling the distance constraints were matched to the structural model and connected different parts of the ATPase domain and the SAM-key (Fig 4C), which included the interaction surface of protrusion I and SAM-key (Fig 4C). The distribution of crosslinking length as mapped to the model shows mostly crosslinks fulfilling the distance constraint (Fig S4D). Overall similar results were obtained when the SAM-key was crosslinked to Fun30ΔSAM (Fig S4A–D), even though high concentrations of the isolated SAM-key may also lead to inter-protein crosslinks between different SAM-key molecules. The validated in silico structural model therefore indicates that the SAM-key domain would contact a part of the ATPase domain that is known to facilitate regulation of catalytic activity in nucleosome remodelers (Szerlong et al, 2008; Clapier et al, 2016, 2020; Xia et al, 2016; Zhou et al, 2016; Liu et al, 2017; Eustermann et al, 2018; Knoll et al, 2018; Li et al, 2019).

### Structural models of Fun30 in complex with nucleosomes suggests similarities to other remodelers

Protrusion I is an extension of the N-terminal RecA-like lobe of Swi2/Snf2-type nucleic acid translocases, which in INO80/SWR1 and SWI/SNF remodelers interacts with the post-HSA domain (Szerlong et al, 2008; Clapier et al, 2016, 2020; Xia et al, 2016; Liu et al, 2017; Eustermann et al, 2018; Knoll et al, 2018; Li et al, 2019; Jungblut et al, 2020). Pioneering work on the RSC complex has shown that a key function of the protrusion I–post-HSA interaction is to promote coupling of ATP hydrolysis and DNA translocation (Clapier et al, 2016). The post-HSA domain and preceding HSA domain are proposed to couple motor/remodeling activity to substrate recognition and sense extranucleosomal linker DNA or other regulatory input in different remodelers (Eustermann et al, 2018; Knoll et al, 2018; Turegun et al, 2018; Baker et al, 2021; Kunert et al, 2022). The direct interaction of the Fun30 SAM-key domain with protrusion I suggests that the SAM-key might fulfill a related regulatory function for Fun30-remodeling activity and prompted us to compare the Fun30 model with the structures of various remodelers containing post-HSA domains or helical regulatory elements at protrusion I.

---

of ATP, but not ATPγS, allows sliding. Representative gel of n = 4 biological replicates. **(C)** Sliding assay as in (B), but with Fun30 WT, Fun30-K603R (Walker A mutant), and Fun30ΔSAM proteins. Representative gel of n = 4 biological replicates. **(D, E, F, G)** The SAM-key is required for nucleosome eviction. **(D)** Schematic of nucleosome eviction assay: addition of remodeler, ATP, and histone chaperone Nap1, which acts as an acceptor for histone H2A–H2B dimers and H3–H4 tetramers, allowing to monitor eviction. In addition, because end-positioned nucleosomes are used, also sliding towards the center of the DNA fragment can be observed. **(E)** Eviction of nucleosomes with labelled H2A (46-C-D550). Eviction is seen by (i) decrease of labeled nucleosome (top), (ii) decrease of nucleosome signal in ethidium bromide stain (second from top), (iii) increased Nap1-bound labelled histone (third from top), (iv) increase of "free" DNA in ethidium bromide stain (bottom). ATP- and remodeler-dependent eviction is shown by the addition of ATP, ATPγS, and Fun30. Representative gel of n = 3 biological replicates. **(F)** Eviction assay as in (E), but with Fun30 WT, Fun30-K603R (Walker A mutant), and Fun30ΔSAM mutant and with labeled H2A (46-C-D550, left) and labeled H4 (64-C-D550, right). Representative gels of n = 4 biological replicates. **(G)** Quantification of nucleosome eviction of Fun30 WT (dark blue) or Fun30ΔSAM (light blue) by free DNA signal (H4-label as representative); shown is intensity of free DNA peak normalized to control without remodeler. n = 4 replicates, filled circles indicate the replicates, thick lines the mean, error-bars depict SD. Dark blue = Fun30 WT, light blue = Fun30ΔSAM. **(H, I)** The SAM-key is required for DNA-stimulated ATPase activity of Fun30. **(H)** Schematic of DNA-stimulated ATPase activity: nucleosome remodeler can be stimulated to hydrolyze ATP when in the presence of DNA or nucleosomes as stimulus. **(I)** Colorimetric ATPase assay using Fun30 (dark blue), Fun30ΔSAM (light blue), ATP and DNA stimulus (herring sperm DNA, 100 ng/µl). Turnover rate $k_{cat}$ was calculated as the number of ATP molecules hydrolyzed per second per remodeler enzyme under conditions of enzyme saturation. n = 4 replicates shown is mean, error-bars depict SD. **(J, K)** The isolated SAM-key can complement defects of Fun30ΔSAM when added in trans. **(J)** Colorimetric ATPase assay using Fun30 (dark blue), Fun30ΔSAM (light blue), ATP and DNA stimulus (herring sperm DNA 100 ng/µl) and titrating different amounts of the SAM-key in trans. Near WT levels of ATP hydrolysis can be observed ($k_{cat}$ ~2.0) when excess (500x) of the SAM-key was added to Fun30ΔSAM. The mean of n = 2 biological replicates is shown; error-bars depict SD. **(K)** Sliding of nucleosomes with labelled H4 (64C-D550) assembled on a 100W0 fragment (247 bp fragment with end-positioned Widom 601-positioning-sequence) as seen by upshift in gel. Shown is dependence on the SAM-key domain as the Fun30ΔSAM cannot slide but addition of a SAM-key construct (50x molar excess) in trans allows sliding. Representative gel of n = 3 biological replicates.
Source data are available for this figure.

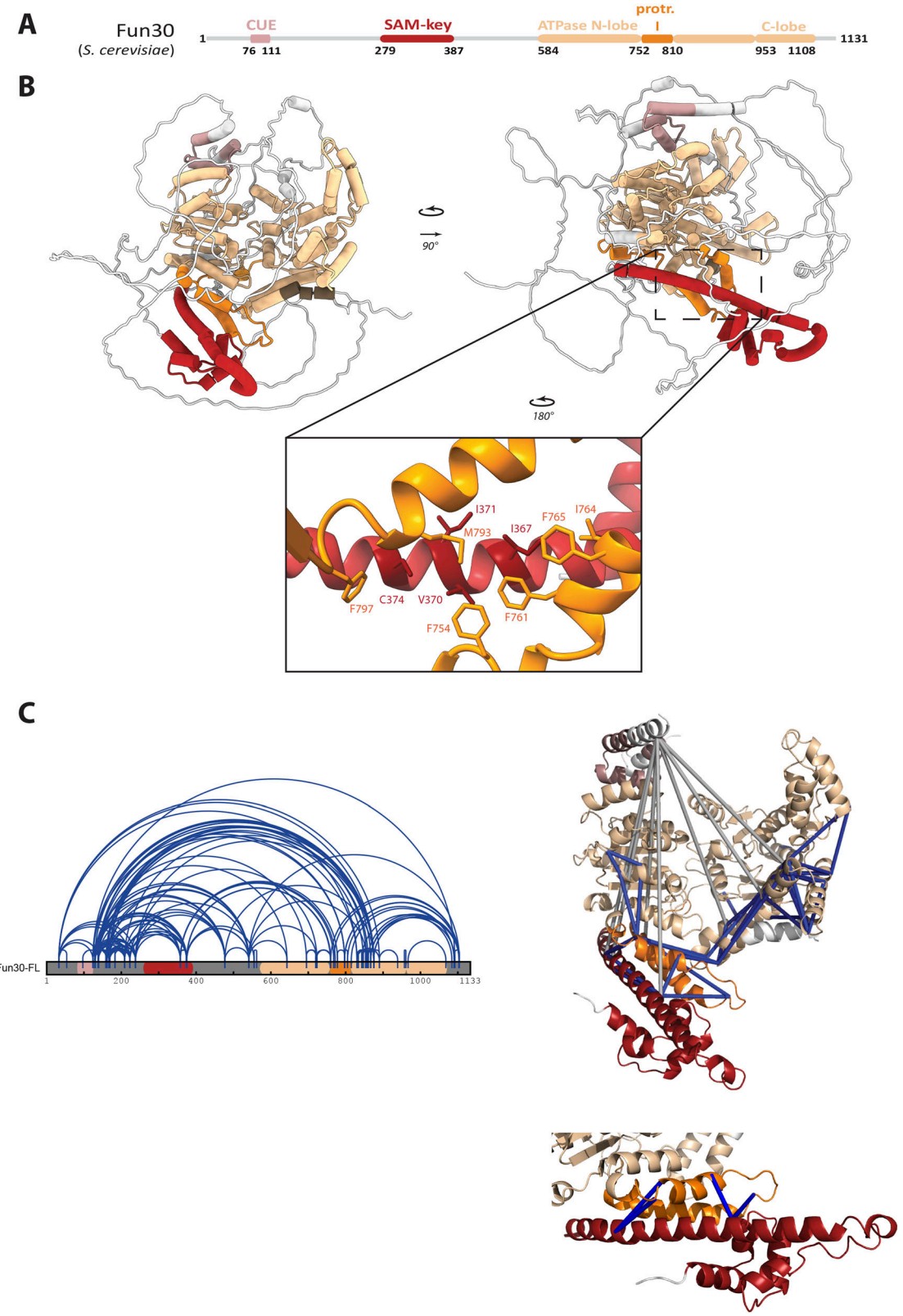

**Figure 4. XL-MS verifies AlphaFold2 model of Fun30 and contacts between the SAM-key and protrusion I located within the SNF2-ATPase domain.**
**(A)** Fun30 domain architecture with CUE domain (rosy brown), SAM-key (red) and SNF2-type ATPase domain consisting of N-terminal lobe (beige), protrusion I (orange), insertion II (beige), and C-terminal lobe (beige). Color scheme used throughout figure. **(B)** AlpaFold2 modelling of Fun30 with high confidence in predicted structured regions (colored). Box indicates the region of zoom in: SAM-key (red) with predicted interaction to the protrusion I (orange). Amino acids likely contributing to the

Different lines of evidence suggested that nucleosome remodelers may interact with nucleosomes at defined yet different sites, including super helical locations SHL+2, SHL−6, and the dyad axis (reviewed in Morgan et al [2021]). In particular, recent cryo-EM analysis showed that human SMARCAD1 engages in an unusual contact with nucleosomes at the dyad axis, but the overall resolution was too low to unambiguously dock the Fun30 ATPase and especially to detect the SAM-key (Markert et al, 2021). To understand how (i) Fun30 may engage a nucleosome, (ii) reveal how the SAM-key may be placed in relation to nucleosomal DNA, and (iii) interrogate whether it resembles elements in other nucleosome remodelers, we took a broader view and aligned the Fun30 AlphaFold2 model with high-resolution remodeler structures engaging the nucleosome at SHL+2 and SHL−6 (Figs 5A–C and S5). We also docked the Fun30 model to the dyad, using the typical Swi2/Snf2 ATPase:DNA interactions at SHL+2 as guide. Structural alignment of the Fun30 model with Sth1 ATPase (RSC complex) (Du et al, 1998), which engages the nucleosome at SHL+2, shows that the predicted conformation of the SAM-key resembles the conformation and mode of interaction of the Sth1 post-HSA domain with protrusion I (Fig S5B and C). Alignment of the Fun30 model with the Ino80 ATPase at SHL−6 (Kunert et al, 2022, Figs 5B and S5B and C) reveals that in this model, the long helix of the SAM-key projects along DNA at the entry site in a manner remarkably similar to the INO80 post-HSA/HSA domain (Fig 5A). In both docking models, the SAM-key is located close to (extra-)nucleosomal DNA, without causing structural clashes. Docking at SHL 0 (dyad) indicates that at this location, the SAM-key may contact nucleosomal DNA (or a bound protein) close to the entry site (Figs 5C and S5A and C).

### The protrusion I–SAM-key interface is required for Fun30-remodeling activity

The apparent similarity to HSA/post-HSA domains of Ino80 and Sth1 suggests that the SAM-key helix may be (at least in part) a structural and functional analog. We observed a cluster of positively charged amino acids (KRKRR 338–342) in a loop at the tip of the SAM-key which according to the structural models may be poised for interaction with DNA. However, these residues do not appear to be highly conserved in Fun30-SMARCAD1-ETL remodelers (Fig S1B). To ascertain the functional importance of SAM-key:protrusion I and putative SAM-key:DNA interactions, we tested mutant proteins in functional assays. To test the basic residues that could be involved in DNA binding, we deleted the positively charged amino acids (ΔKRKRR), but the Fun30ΔKRKRR protein retained DNA-stimulated ATPase and remodeling activities (Fig S6A–D). These data indicate that the basic amino acids on the tip of the SAM-key are either not involved in DNA binding, that this protein–DNA

interaction is not important for Fun30 functions tested here, or that other parts of the protein function redundantly.

Next, to test the importance of the interaction surface between SAM-key and protrusion I, we mutated two hydrophobic amino acids (I367, C374) to charged, bulky arginine residues (Fun30-ICRR) to weaken or abolish this inter-domain interaction. Purified Fun30-ICRR was still able to bind to DNA and nucleosomes similarly as the WT protein (Figs 6A–D and S6E and F), suggesting the protrusion I–SAM-key interaction is not involved in nucleosome binding, as predicted. When we tested nucleosome remodeling, however, we found that even at high concentrations, Fun30-ICRR was neither able to slide nor evict nucleosomes (Fig 6E and F). These data further verified the structural model from Fig 4, and we conclude that nucleosome-remodeling activity by Fun30 is abrogated by the disruption of SAM-key binding to protrusion I.

Given the known role of protrusion I in regulating ATPase activity of remodelers, we also tested whether Fun30-ICRR was ATPase active upon DNA stimulation (Fig 6G). Here, we observed that Fun30-ICRR showed a strong defect in DNA-stimulated ATPase activity, similar to Fun30ΔSAM (Fig S6G). This defect could, however, be complemented by the addition of the isolated SAM-key (Fig S6G) as could the defect in nucleosome sliding (Fig S6H), suggesting that extrinsically added SAM-key can interact with protrusion I within the context of the Fun30-ICRR protein and restore function. In line with published work on other remodelers (Szerlong et al, 2008; Clapier et al, 2016, 2020; Li et al, 2019; Jungblut et al, 2020), we therefore conclude that protrusion I is a key element of regulation. In Fun30, protrusion I is contacted by the SAM-key, which facilitates allosteric activation of the remodeler.

## Discussion

Despite many nucleosome remodelers being complex molecular machines consisting of multiple subunits, the existence of single-subunit remodelers suggests that only few elements are necessary in addition to the SNF2-ATPase domain to catalyze the principal nucleosome-remodeling reaction. Here, we identify one such mechanism for the Fun30-SMARCAD1-ETL subfamily of remodelers. This mechanism involves a protein domain that we annotate SAM-key, which our analysis indicates can bind to protrusion I within the SNF2-ATPase domain. Although the SAM-key is not required to bind to DNA or nucleosomes, it is required for catalytic activity of Fun30. In particular, DNA fails to stimulate ATPase hydrolysis by the remodeler in the absence of the SAM-key. This suggests a model, whereby the SAM-key mediates allosteric activation of nucleosome remodeling.

SAM domains are found in several proteins and mediate various functions, from protein interaction to RNA and DNA binding (Kim & Bowie, 2003). This suggests that the actual SAM-like fold functions

hydrophobic interaction surface are highlighted: I367, V370, I371, and C374 of the SAM-key (red)—F754, F761, I764, F765, M793, F797 of protrusion I (orange). **(C)** XL-MS with BS3 crosslinking verifies the AlphaFold2 model. Left: 2D representation of crosslinks in blue on Fun30 (n = 153, unfiltered). Right: 3D-mapping of crosslinks on AlphaFold2 model. Crosslinks in low confidence, unstructured regions were omitted, leaving crosslinks within predicted structured regions ± two additional aa residues were considered (n = 47). Blue crosslinks (n = 40) match the model with a length restriction for BS3 of 35 Å. Grey crosslinks (n = 7) violate the threshold and are >35 Å, but all involved the CUE domain which might be wrongly positioned in the model or dynamic. Zoom in: SAM-key showed four crosslinks, all matching 35 Å distance constraint and confirming the position in close proximity to protrusion I.

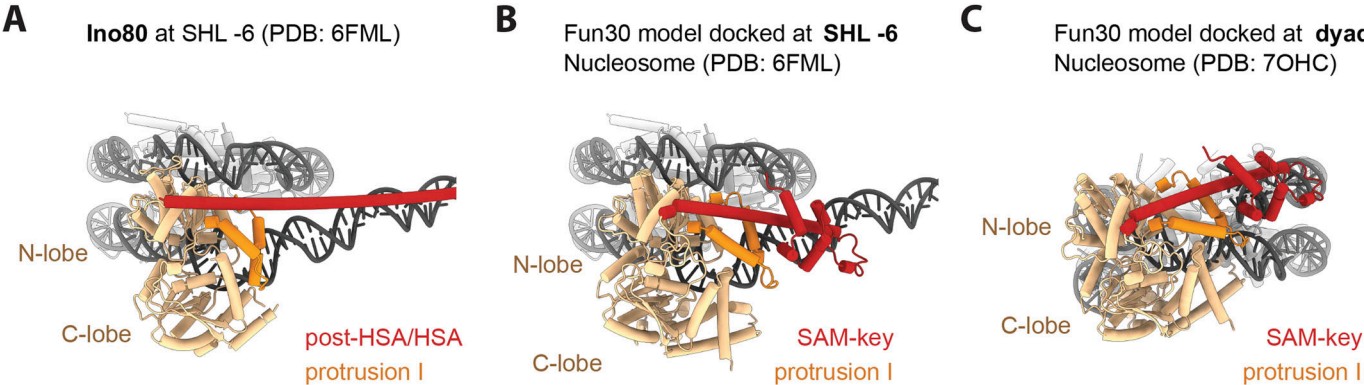

**Figure 5. Comparison of the Fun30 model with structures of the chromatin remodelers INO80 and RSC and docking of the model to a nucleosome structure accordingly.**
**(A)** Structure of a Ino80 bound to nucleosome at SHL–6 in the context of the nucleosome (PDB:6FML, Kunert et al, 2022) shown for comparison. The ATPase N- and C-lobes (beige), protrusion I (orange), and the post-HSA/HSA domains (red) are color-coded. **(B)** Fun30 model docked at SHL–6. The Fun30 model obtained from the AlphaFold database was aligned with the structure of the Ino80 ATPase bound at SHL–6 (model based on PDB:6FML). The ATPase N- and C-lobes (beige), protrusion I (orange) and the SAM-key (red) are color-coded. **(C)** Fun30 model docked at the dyad. Docking to the dyad of a nucleosome (PDB: 7OHC) was guided by the structure of Sth1 bound at SHL+2 (PDB: 6TDA). Color coding as above.

to position binding surfaces in the right configuration. In case of Fun30, the structural model suggests that this binding surface is formed by the "key" helix, which interacts with protrusion I of the ATPase. This interaction appears to be analogous to the interaction of the post-HSA helix with protrusion I in Ino80, Snf2 or Sth1 (Liu et al, 2017; Clapier et al, 2020; Han et al, 2020; Wagner et al, 2020; Baker et al, 2021; Kunert et al, 2022).

In the case of INO80, the post-HSA is connected to the HSA domain, which recruits actin and actin-related proteins, forming a regulatory domain that interacts with extranucleosomal DNA (Ayala et al, 2018; Brahma et al, 2018; Eustermann et al, 2018; Knoll et al, 2018). Mutating the HSA domain does not kill the ATPase activity of INO80, but rather decouples ATP hydrolysis from nucleosome sliding, suggesting that protrusion I interactions with regulatory elements are key to transduce signals that control the activity of the remodeler. Whether this takes place also in Fun30 needs to be addressed in future studies, but the presence of a SAM-like domain at the end of the key regulatory helix strongly suggests a functional interplay between Fun30 activity and SAM-mediated macromolecular interactions.

Protrusion I has already emerged as a key element controlling ATP hydrolysis and/or motor activity in many remodelers (RSC, SWI/SNF, Snf2, ISWI) (Szerlong et al, 2008; Clapier et al, 2016, 2020). The fact that protrusion I binds to other parts of these remodelers such as auto-N (ISWI) or post-HSA (RSC, SWI/SNF, and INO80) strongly suggests the existence of a rather widely conserved allosteric mechanism controlling ATP hydrolysis. Recent data showed that in RSC, post-HSA might adopt different conformations (Baker et al, 2021), which may be part of a regulatory or even mechanical cycle, but how the allosteric activation works in detail needs future structural work at higher resolution and involving different functional nucleotide states of remodelers, which are still scarce. ATPase activity of Fun30 and other remodelers has been shown to be stimulated by DNA and nucleosome binding (Laurent et al, 1993; Cairns et al, 1996; Corona et al, 1999; Tran et al, 2000; Awad et al, 2010; Adkins et al, 2017), and we now show that the SAM-key and its interaction with protrusion I is required for

Fun30 catalytic activity, suggesting that upon DNA binding, the SAM-key mediates allosteric activation of the remodeler. This is further supported by our observation that the SAM-key can rescue the Fun30ΔSAM and Fun30-ICRR mutants when added in excess in trans. In future, structural work will be required to test whether the SAM-key–protrusion I interaction surface may be conformationally flexible and how it is precisely positioned relative to the nucleosome and nucleosomal DNA.

The identity of Fun30's stimulus and how it is transmitted to the ATPase is a matter of an ongoing debate (Awad et al, 2010; Adkins et al, 2017). ATP hydrolysis by certain remodelers was shown to be strongly stimulated by nucleosomes, but more poorly by DNA, suggesting stimulation by the substrate (Hauk et al, 2010; Mueller-Planitz et al, 2013). Similar to what was found for RSC, Fun30 can be stimulated very efficiently by DNA alone (Boyer et al, 2000; Saha et al, 2002). Furthermore, single-stranded DNA (120 nt) or sheared herring sperm DNA are efficient stimuli, and we note that shearing may generate ssDNA or ss-dsDNA junctions. Furthermore, the ssDNA constructs used in the assay may form secondary structures and generate ss-dsDNA junctions. Taking into account that Fun30 works to promote DNA end resection at DSBs and that our previous work localized Fun30 to sites of ss-ds-DNA junctions where it is recruited by the 9-1-1 complex (Bantele et al, 2017; Bantele & Pfander, 2019), it is tempting to speculate ss-ds-DNA junctions or single-stranded DNA are bound by Fun30 and may stimulate ATP hydrolysis, possibly involving the SAM-key. To test this model, we will require to not only use different DNA substrates compared with what we have done here, but also to model damaged chromatin including the ssDNA-binding protein RPA, nucleosomes, and to include proteins such as 9-1-1 and Dpb11, which target Fun30 to damaged chromatin. Lastly, previous work by us and others showed that Fun30 and SMARCAD1 are targets of post-translational modification, particularly CDK phosphorylation (Chen et al, 2016; Bantele et al, 2017). Although our previous investigation of CDK phosphorylation-defective Fun30 mutants showed defects already in recruitment of the remodeler to DNA damage sites (Bantele et al,

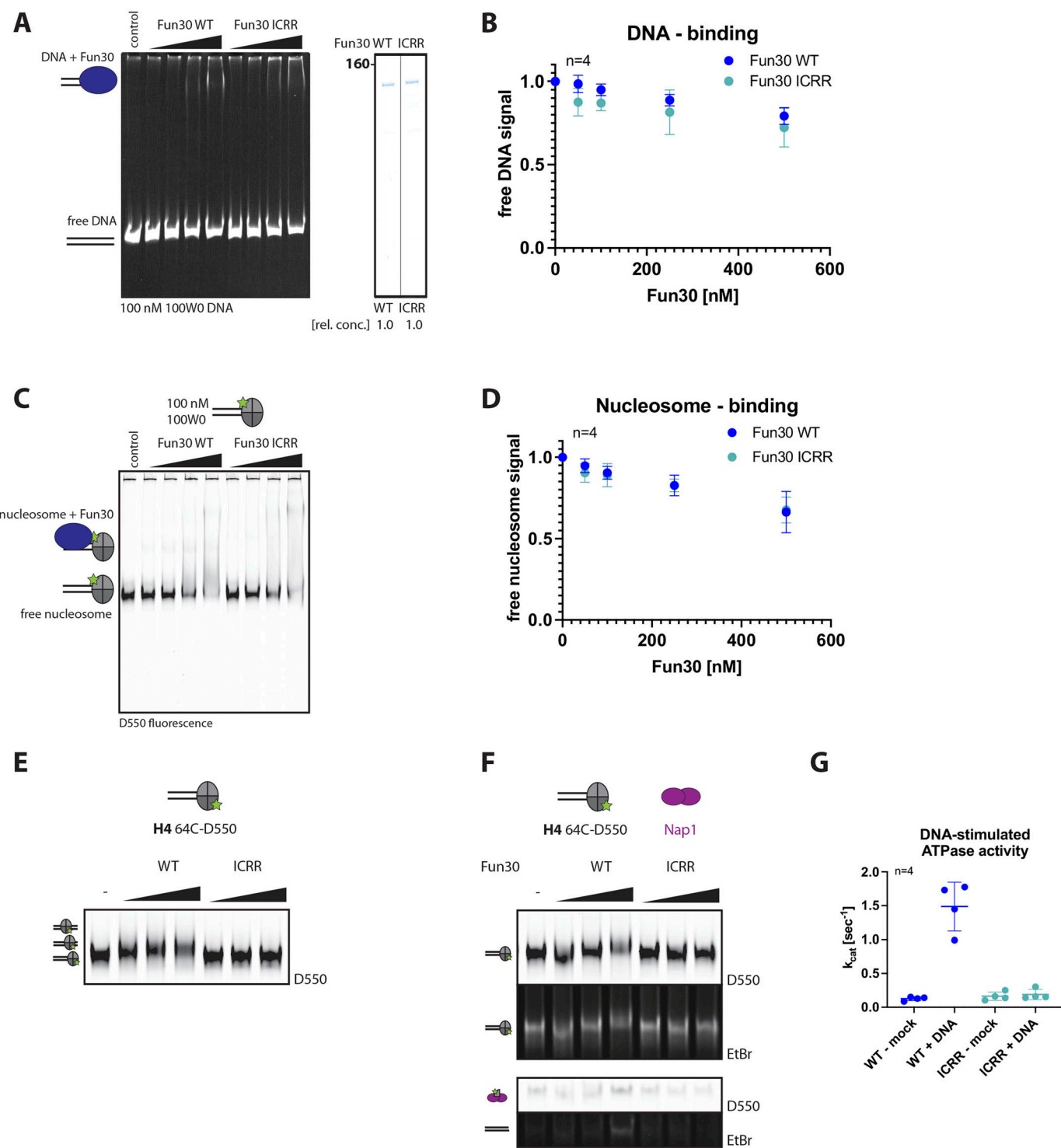

**Figure 6. The SAM-key interaction with the ATPase domain is essential for nucleosome remodeling by Fun30.**
**(A, B, C, D)** A mutant (Fun30 I367R, C374R [Fun30-ICRR]) with a defect in the SAM-key protrusion I interface binds normally to DNA and nucleosome, and behaves therefore similar to Fun30ΔSAM. **(A)** DNA binding of Fun30-ICRR and quantification in gel shifts. DNA binding to a 247-bp dsDNA construct. Left: gel shift in native gel with Fun30 WT and Fun30-ICRR and DNA stained with ethidium bromide. Representative gel of n = 4 biological replicates. Right: Coomassie staining shows equal amounts of WT and mutant protein were used in both DNA- (A) and nucleosome-binding experiments (C) (quantified from band intensity). **(B)** Quantification of DNA binding as in Fig 2A. n = 4 replicates, filled circles indicate mean, error-bars depict SD. Individual datapoints of replicates are shown in Fig S6E. **(C, D)** Nucleosome binding by Fun30-ICRR and quantification. **(C)** Binding to 100W0 nucleosomes of Fun30 WT and Fun30-ICRR. Gel shift shown by fluorescence imaging of the labeled histone H2A (46-C-D550) in native gel. Representative gel of n = 4 biological replicates. **(D)** Quantification of nucleosome binding as in Fig 2B. n = 4 replicates, filled circles indicate mean, error-bars depict SD. Individual datapoints of replicates are shown in Fig S6F. **(E, F, G)** Fun30 nucleosome remodeling requires the SAM-key interaction with protrusion I of the ATPase

2017), we can at this point not exclude that phosphorylation or complex formation also have an additional role in activating the remodeler in situ at DNA damage sites.

What is the relevant catalytic activity of Fun30? Recent work of the Luger laboratory indicated that human SMARCAD1 may follow an unconventional mechanism where it binds to nucleosomes at the dyad and evicts entire nucleosomes (Markert et al, 2021). Our work suggests that Fun30 is able to evict both H2A–H2B dimers and H3–H4 tetramers when Nap1 is present suggesting that also Fun30 may lead to nucleosome eviction. At the same time, Fun30 is also able to slide nucleosomes similar to other remodelers. Whether Fun30 and SMARCAD1 act as nucleosome evictors at DSBs in vivo is difficult to ascertain for two reasons. Eviction and resection appear to be intrinsically coupled. Therefore, although a *fun30* mutant strain showed reduced eviction, it is unclear whether that is simply a secondary defect arising from a primary resection defect (Peritore et al, 2021). Second, SWI/SNF and RSC appear to be major evictors at DSBs (Peritore et al, 2021). Therefore, and because many *fun30* phenotypes can be suppressed by mutation of the resection inhibitor Rad9 (Chen et al, 2012; Bantele et al, 2017), we prefer a model whereby Fun30-SMARCAD1-ETL affect damaged chromatin more specifically and act on nucleosomes that are bound by Rad9-53BP1 (Bantele & Pfander, 2019; Karl et al, 2022).

Drugs targeting the DNA damage response are becoming increasingly important in cancer therapy (Li et al, 2020) as are those targeting chromatin factors (Kaur et al, 2019). Highly conserved SNF2-ATPase domains, however, bring about the problem that active site inhibitors may lack specificity (Dutta et al, 2012; Rakesh et al, 2021). Our finding that the SAM-key–protrusion I interaction surface is fully required for the catalytic activity of Fun30 and that addition of the SAM-key in trans can interact with the remodeler raises the possibility that targeting this interaction surface may be an alternative strategy for developing inhibitors to SMARCAD1 and other remodelers. Given the unique nature of the modules (SAM-key, post-HSA, others) binding to protrusion I, we speculate that it may be suited for the development of compounds with superior specificity.

# Materials and Methods

## Yeast cultivation, strains, plasmids, and antibodies

All yeast strains used in this study are derived from W303 MATa (strains listed in Table S1, Rothstein, 1983) and were constructed using PCR-based tagging or deletion of yeast genes (Knop et al, 1999). Cells were grown in YP-glucose or YP-raffinose media at 30°C. Cell cycle synchronization in M-phase was performed using nocodazole for 2–3 h and controlled by flow cytometry. For spot assays, pre-cultures were grown to the stationary phase overnight and a serial dilution series (1:5) was spotted on respective selective/drug-containing plates and YPD plates. For survival assays on CPT, a seven-step serial dilution series (1:5 dilution) was prepared starting at $OD_{600}$ 1.0 and spotted on YPD plates with different concentrations of CPT (6, 10 or 12 $\mu$g/ml). For silencing assays, a six-step serial dilution series (1:5 dilution) was prepared starting at $OD_{600}$ 0.5 and spotted on YPD-, SC-Ura-, and 5'-FOA-plates.

For molecular cloning, genes were amplified from yeast genomic DNA and inserted in plasmids using the In-Fusion HD cloning kit (Clontech). For site-directed mutagenesis, a PCR-based protocol with mutagenic oligonucleotides was used. All plasmids used in this study are listed in Table S2 and all antibodies in Table S3.

## Chromatin immunoprecipitation (ChIP)

As proxy for DNA-end resection, the ssDNA was purified by chromatin immunoprecipitation of RPA. Therefore, cells were grown in YP raffinose to an $OD_{600}$ of 0.5 and cell cycle arrest in the M phase was induced using nocodazole (5 $\mu$g/ml). Arrests were confirmed using a microscope. A DSB at the MAT locus was introduced by HO endonuclease expressed from pGAL1-10 promoter by addition of galactose (final concentration 2%). 100 ml samples were crosslinked with formaldehyde (final concentration 1%) for 16 min at indicated timepoints and the reaction was quenched with glycine (final concentration 450 mM). Cells were harvested by centrifugation, washed in ice-cold PBS, and snap-frozen. For lysis, cell pellets were resuspended in 800 $\mu$l lysis buffer (50 mM HEPES KOH pH 7.5, 150 mM NaCl, 1 mM EDTA, 1% Triton X-100, 0.1% Na-deoxycolate, 0.1% SDS) and grinded with zirconia beads using a bead-beating device (MM301; Retsch). The chromatin was sonified to shear the DNA to a size of 200–500 bp using Bioruptor (Diagenode). Subsequently, the extracts were cleared by centrifugation, 1% was taken as input sample, and 40% were incubated for 90 min with anti RFA antibody (AS07-214; Agrisera) followed by 30 min with Dynabeads ProteinA (Invitrogen). Beads were washed 3x in lysis buffer, 2x in lysis buffer with 500 mM NaCl, 2x in wash buffer (10 mM Tris–Cl pH 8.0, 0.25 M LiCl, 1 mM EDTA, 0.5% NP-40, 0.5% Na-deoxycholate), and 2x in TE pH 8.0. DNA–protein complexes were eluted in 1% SDS, proteins were removed with Proteinase K (3 h, 42°C) and crosslinks were reversed overnight at 65°C. The DNA was subsequently purified using phenol–chloroform extraction and ethanol precipitation and quantified by quantitative PCR (Roche Light-Cycler480 System, KAPA SYBR FAST 2x qpCR Master Mix, KAPA Biosystems) at indicated positions with respect to the DNA DSB. As control, 2–3 control regions on other chromosomes were quantified.

## Recombinant proteins

### Fun30

A plasmid harbouring the respective Fun30 construct (e.g., pLAK080 for Fun30 WT) with N-terminal 6xHis-GST-3C-cleavage site was

---

catalytic domain. **(E)** Nucleosome-sliding assay (with 100W0 end-positioned nucleosomes and labelled H4 as in Fig 3B) shows the nucleosome-sliding defect of Fun30-ICRR. Representative gel of n = 2 biological replicates and n = 4 technical replicates. **(F)** Eviction assay (labelled H4 as in Fig 3F) shows nucleosome eviction defect of Fun30-ICRR. Representative gel of n = 4 biological replicates. **(G)** DNA-stimulated ATPase assay as in Fig 3I but with Fun30-ICRR. Unlike Fun30 WT that reaches a $k_{cat}$ of 1.5 s$^{-1}$, Fun30 ICRR only reaches a $k_{cat}$ of 0.2 s$^{-1}$ comparable with Fun30ΔSAM. n = 4 replicates shown by filled circles, central line indicates mean, error-bars depict SD. Source data are available for this figure.

transformed into *E. coli* BL21 DE3 pRIL. Cells were cultivated at 37°C, 220 rpm (Innova 44, New Brunswick) in double-selective LB-medium (100 μg/ml ampicillin [Amp] and 34 μg/ml chloramphenicol [Chl]) to an $OD_{600}$ of ~1.0. Addition of IPTG (1 mM final, 2316.4; *Roth*) induced overexpression of the construct, which was performed overnight at 18°C. Cells were harvested by centrifugation, washed in ice-cold PBS, and snap-frozen or directly processed. Unless specified, all further steps were performed on ice/at 4°C. Cells were lysed in lysis buffer (50 mM HEPES KOH pH 7.5, 500 mM NaCl, 10% glycerol, 0.5 mM CHAPS, 2 mM ß-mercaptoethanol, 1x cOmplete protease inhibitor cocktail EDTA-free (Roche), and 10 μg/ml leupeptin, 1 μg/ml pepstatinA, 1 mM benzamidine, 2 μg/ml aprotinin, 1 mM AESBF) with a combination of lysozyme (1 mg/ml) and sonication (3 × 5 min, 2 s on, 2 s off; Bandelin *Sonopuls UW 2070*). Lysate was cleared with SmDNase (750 U/ml lysate) and centrifugation. Cleared lysate was incubated with Ni-NTA-agarose (1 ml bed volume/L culture; QIAGEN) for 1 h. Beads were washed (lysis buffer) and proteins eluted (lysis buffer + 1 M imidazole). Eluate was diluted (100 mM imidazole final), incubated with glutathione sepharose 4 FF (1.5 ml bed volume/L culture; Cytiva) for 2 h. Beads were washed (lysis buffer) and protein eluted by cleaving off the tags using His-3C-protease (lysis buffer + 17 U/ml His-3C [homemade]). Eluate was concentrated to 500 μl (Amicon Ultra 4, 10,000 MWCO) and run on superdex 200 size exclusion column (S200 Increase 10/300 GL, 24 ml column volume; Cytiva), 500 μl fractions were collected and the fractions analyzed by SDS–PAGE and Coomassie staining. The fractions were aliquoted, snap-frozen, and stored at −80°C. Fun30-3xFLAG-CBP was purified from yeast as described (Bantele et al, 2017).

### Nap1

A plasmid harbouring the respective Nap1 construct (pCFK1 [Kurat et al, 2017]) with N-terminal GST-3C-cleavage site was transformed into *E. coli* BL21DE3 pRIL. The cells were grown at 37°C, 220 rpm (Innova 44, New Brunswick) in double-selective LB-medium (100 μg/ml ampicillin [Amp], and 34 μg/ml chloramphenicol [Chl]) to an $OD_{600}$ of ~1.0. Addition of IPTG (1 mM final) induced overexpression of the construct, which was performed for 2 h at 37°C. Cells were harvested by centrifugation, washed in ice-cold PBS, and snap-frozen. Unless specified, all further steps were performed on ice at 4°C. Cell pellets were lysed in Nap1 lysis buffer (100 mM $KxPO_4$ pH 7.6, 150 mM KOAc, 5 mM $MgCl_2$, 0.5 mM CHAPS, 1 mM DTT, 1x cOmplete protease inhibitor cocktail EDTA-free [Roche], and 10 μg/ml leupeptin, 1 μg/ml pepstatinA, 1 mM benzamidine, 2 μg/ml aprotinin, 1 mM AESBF) with a combination of lysozyme (1 mg/ml) and sonication (3 × 5 min). Lysate was cleared with SmDNase and centrifugation. Cleared lysate was incubated with glutathione sepharose 4 FF (1.5 ml bed volume/L culture; Cytiva) for 2 h. Beads were washed (lysis buffer) and protein eluted by cleaving off the tags using His-3C-protease (lysis buffer + 17 U/ml His-3C [homemade]). Eluate was dialyzed for 2 h (3,500 MWCO, G2 cassette, Slide-a-Lyzer, Thermo Fisher Scientific) with dialysis buffer (20 mM Tris pH 7.5, 100 mM NaCl, 0.5 mM EDTA, 10% glycerol, 1 mM DTT, 0.1 mM PMSF). Nap1 was further purified using MonoQ column (Cytiva) using a 20 CV gradient from 0.1 to 1 M NaCl (20 mM Tris–HCl ph 7.5, 100 mM NaCl, 0.5 mM EDTA, 10% glycerol, and 1 mM DTT) and the fractions analyzed by SDS–PAGE and Coomassie staining.

### Histone expression, purification and labeling, and nucleosome assembly

Genes encoding WT *Saccharomyces cerevisiae* histones were codon optimized and synthesized (Genscript) for bacterial expression. H2A, H2B genes were cloned into pETDuet and H3, H4 were cloned into pCDFDuet vectors (#71146, #71340; Novagen). The mutants H2A_46C and H4_64C were generated using QuickChange mutagenesis (#200515; Agilent). Combination of two vectors pETDuet_H2A_46C-H2B + pCDFDuet_H3-H4 and pETDuet_H2A-H2B + pCDFDuet_H3-H4_64C were co-transformed in *E. coli* BL21 DE3 codon plus pRIL (Agilent) and grown in ZYP-5052 auto-induction media (Studier, 2005) at 37°C up to OD600 = 0.8. The temperature was lowered to 18°C and expression continued further for 16 h. All subsequent steps were performed at 4°C. The cells were harvested by centrifugation (4,000*g*, 15 min), resuspended in buffer A (20 mM HEPES-NaOH, pH 7.6, 10% glycerol, 1 mM EDTA) + 0.8 M NaCl, 1 mM DTT, supplemented with 1 vial of protease inhibitor cocktail (#39102.03; Serva) and lysed by sonication. The cell lysate was cleared by centrifugation (23,666*g*, 45 min) and applied to 2x HiTrap Heparin HP (#17040701; Cytiva) 5 ml columns equilibrated in buffer A + 0.8 M NaCl, 1 mM DTT. Columns were washed with 10 CV buffer A + 0.8 M NaCl, 1 mM DTT, and histone octamers were eluted with a 0.8–2 M NaCl linear gradient. Peak fractions were pooled and spin concentrated (Amicon Ultra, MWCO 10,000 #UFC901024; Merck). The concentrated protein complex was applied to a HiPrep 26/10 (#17508701; Cytiva) desalting column equilibrated with buffer A to remove DTT, peak fractions were collected, and concentration was measured. DyLight 550 Maleimide (#62290; Thermo Fisher Scientific) was added to the protein in 20-fold molar access. The reaction was allowed to proceed over night at 4°C protected from light. Upon completion of the reaction, the conjugate and free dye were separated on a Superdex 200 increase 10/300 GL (#28990944; Cytiva) size exclusion column equilibrated in buffer A + 2 M NaCl, 1 mM DTT. Peak fractions containing histone octamers were pooled, spin concentrated, frozen in aliquots in liquid nitrogen, and stored at −80°C.

DNA was used in nucleosomes containing the 147-bp Widom 601-nucleosome-positioning sequence (Lowary & Widom, 1998) at the end of the sequence. A 100-bp overhang on one side was used to generate 100W0 nucleosomes. Large-scale PCR amplification of 100W0 from plasmid pLAK148 was performed using PCR-based strategy and PCR products were purified using a 1 ml HiTrap Q HP column (Cytiva). DNA was eluted with a gradient from 100% buffer A (TE + 50 mM NaCl) to 100% buffer B (TE + 1 M NaCl) over 20 column volumes. Fractions containing the DNA were pooled, subjected to ethanol precipitation, and finally resuspended in HE buffer (10 mM HEPES-KOH pH 7.6, 1 mM EDTA), then stored at −20°C.

For nucleosome assembly, the established protocol (Luger et al, 1999; Dyer et al, 2004) was slightly adapted. In short, dialysis buttons (3,500 MWCO, Slide-A-Lyzer Mini dialysis unit; Thermo Fisher Scientific) were prepared and equilibrated according to the manufacturer's instructions.

Nucleosome assembly reactions were combined from the 100W0-DNA, histone octamers, and a 5 M NaCl stock solution. The ratio of octamer:DNA was titrated for optimal assembly for the differently labeled histone octamers, final concentration of DNA and octamer was 2–4 μM. DNA and NaCl (final 2 M) were mixed first,

then the volume was adjusted with HE buffer and the histone octamer was added last.

After mixing, the reaction was transferred to the dialysis buttons and dialysis in RB-high (10 mM HEPES KOH pH 7.6, 1 mM EDTA, 2 M NaCl, 1 mM DTT) was performed at 4°C for 1 h. A setup of peristaltic pumps exchanged RB-high completely with RB-low over 12–16 h, slowly removing dialysis buffer and dripping in the fourfold volume of RB-low (10 mM HEPES KOH pH 7.6, 1 mM EDTA, 100 mM NaCl, 1 mM DTT). After a final dialysis with RB-low over 4 h, the reaction was transferred to low-binding tubes (T4816; Sigma-Aldrich). The efficiency of the assembly was tested by native PAGE, followed by ethidium bromide staining. Nucleosome concentration was estimated using a free DNA control on the gel and calculating the volume of the reaction after dialysis and the amount of free DNA left inside.

## Limited proteolysis

Fun30 proteins were digested with different dilutions of chymotrypsin, elastase, Glu C, subtilisin, and trypsin (Promega). Protein and protease dilutions were made in protease dilution buffer (20 mM HEPES pH 7.5, 50 mM NaCl, 10 mM $MgSO_4$). 1 $\mu$l of diluted protease was added to 600 ng protein in a total volume of 5 $\mu$l. For trypsin, chymotrypsin, and subtilisin the dilutions used were 0.02, 0.005, and 0.001 mg/ml; for GluC and elastase dilutions were 1, 0.1, and 0.01 mg/ml. Cleavage occurred during 30 min incubation on ice. To stop the reaction, 5 $\mu$l 2x Laemmli buffer was added. After 5 min at 95°C, the samples were loaded onto a self-made 10% gel to perform SDS–PAGE followed by silver staining.

## ATPase assay

ATPase activity of Fun30 WT and mutants was analyzed using NADH-coupled assay: an ATP regeneration system (phosphoenolpyruvate, lactate dedydrogenase/pyruvate kinase) replenishes any ATP that is hydrolyzed to ADP + $P_i$, oxidizing one molecule of NADH per regenerated molecule ATP. Measurements of absorption at 340 nm were performed at a *Tecan* infinite M200 Pro plate reader using the i-control 2.0 software in transparent 384-well plates (781186; Greiner BioOne).

For the assay, DNA stimulus (100 ng/$\mu$l herring sperm DNA, 15634-017, for titration of different stimuli 40–1,000 ng/$\mu$l; Invitrogen), NADH (1.5 mM, N8129; Sigma-Aldrich), ATP regeneration system (3 mM phosphoenolpyruvate (10108294001; Roche), 15.5 U/ml lactate dehydrogenase/pyruvate kinase (P0294; Sigma-Aldrich), 10 mM ß-mercaptoethanol), and enzyme (final concentration 100 nM) were mixed in reaction buffer (25 mM HEPES-KOH pH 7.6, 0.1 mM EDTA, 10% glycerol, 100 mM KOAc, 1 mM $MgCl_2$) inside a 384-well plate with a total volume of 30 $\mu$l per well. Reactions were spun down. ATP (R1441; Thermo Fisher Scientific) was added with equimolar $MgCl_2$ (final concentration 1 mM) to start the assay. Before the start, the plate was mixed for 30 s by 300 rpm orbital shaking. Operating temperature of the plate reader was 26°C, kinetic A340 measurements were taken every 10 s within a total of 60 min. ATP consumption was measured in form of NADH decrease. Evaluation of the data was performed with Microsoft Excel: a timeframe from 1,000–2,500 s (or at least 500 s with linear decline of the $A_{340}$-curve) was selected to calculate the slope using the SLOPE function. From the slope value, the turnover rate

$k_{cat}$—the number of ATP molecules hydrolyzed per second per remodeler enzyme—was calculated using the law of Lambert-Beer.

Using the extinction coefficient of NADH 6,220 $M^{-1}$ $cm^{-1}$ and the pathlength of 0.272727 cm for a volume of 30 $\mu$l in one well of the 384-well plate, first, the reaction speed ($v_{max}$) was calculated using $v_{max}$ = slope/(6,220 $M^{-1}$ $cm^{-1}$ × 0.272727 cm). Then, turnover rate $k_{cat}$ [$s^{-1}$] was calculated by dividing $v_{max}$ by the protein concentration used in the assay ($10^{-7}$ M) and correcting by the actual concentration used in the assay obtained from quantification of the input into the ATPase assay from a Coomassie-stained gel with BSA protein standard as described for DNA band quantification below in "DNA and nucleosome-binding assays," using the linear regression of the standard curve for quantification of the protein bands.

To find out the best stimulus for Fun30 ATPase activity, a set of different constructs was tested: Herring sperm DNA (15634-017; Invitrogen), the M13 phage plasmid (NEB), and ss and dsDNA fragments were used as stimuli (see below).

DNA stimuli: Herring sperm DNA (15634-017; Invitrogen).
ssDNA: 120 nt oligonucleotide BP5196 5′CACCTGTTGTAATCGTC TAGAATGGATTATAAAGATGACGATGACAAGGATTATAAAGATGACGATGACA AGGATTATAAAGATGACGATGACAAGATCGAGCTCGAATTCATCGATGAT3′.
M13 ssDNA plasmid (7,249 nt).
dsDNA: 100W0 DNA (247 bp).
25x 601 array DNA 20(W50)$_{25}$20 from pTB127 (4,920 bp).

## In vitro nucleosome-binding assay

For Co-IP experiments, Fun30 constructs with still intact 6xHis-GST-tag or a tag-only construct (final conc. 360 nM) were mixed with H2A-46-C-D550-labeled nucleosome (60 nM) in a total volume of 30 $\mu$l pulldown buffer (50 mM HEPES/KOH pH 7.6, 1 mM EDTA, 150 mM NaCl, 0.1% Tween-20, 10% glycerol, 1 mM DTT, 10 $\mu$g/ml leupeptin, 1 $\mu$g/ml pepstatinA, 1 mM PMSF) and incubated for 30 min on ice. Equilibrated glutathione Sepharose 4 FF (5 $\mu$l bed volume) was added and incubated for 2 h at 4°C with rotation. Supernatant was removed (25 $\mu$l) and mixed with 25 $\mu$l of 2x Laemmli. Beads were washed 3x with 400 $\mu$l buffer. The beads were mixed with equal volume of 2x Laemmli and boiled at 95°C for 5 min. Equal amounts of supernatant and pulldown were loaded on gels and analyzed by fluorescence imaging (Typhoon FLA 9000, GE, in the Cy3-channel for labeled histone H2A) and Western blot (using rabbit anti-H3 primary antibody [1:5,000, ab1791; Abcam] and goat anti-rabbit HRP [1:5,000, 111-035-045; Jackson Immuno Research]).

## Native PAGE gels and gel electrophoresis

To analyze gelshifts with 100W0 DNA or 100W0 nucleosomes, 5% native gels (TBE), equilibrated in 0.2x TBE, and run at 4°C in 0.2x TBE at 180 V for 90 min before imaging fluorescent labels and/or staining with ethidium bromide (1:10,000 in $H_2O$).

## DNA and nucleosome-binding assay

Unless indicated otherwise, all steps were performed on ice or at 4°C. For DNA binding, 100W0 DNA substrate was used. For nucleosome binding, 100W0 nucleosomes were used.

DNA/nucleosomes were diluted (final concentration 100 nM) and mixed in the reaction buffer (15 mM HEPES pH 7.6, 100 mM KOAc, 2 mM MgCl₂, 75 μg/ml BSA, 1 mM DTT). Lastly, respective amount of Fun30 protein was added to a total sample volume of 10 or 15 μl and the reaction incubated for 30 min at 30°C. To check for reversibility, 1 μg of herring sperm DNA was added thereafter and incubation continued for additional 5 min, before loading on native gels. Intensities of free DNA bands were measured using the Fiji distribution of ImageJ (Schindelin et al, 2012; Schneider et al, 2012). In the FIJI software, intensity plots for the entire lane (rectangular selection for control lane, all subsequent lanes selected with an identical rectangle) were made and the peaks corresponding to the respective bands selected for retrieving the integral/area under the curve.

## Nucleosome sliding and eviction assays

Unless specified, all steps were performed on ice. 10 μl reaction volume containing 100 nM 100W0 mononucleosomes with labelled histones (*H2A 46-C-D550* and *H4 64-C-D550*), respective amount of remodeler (5, 25, and 100 nM in titrations) in the reaction buffer (15 mM HEPES pH 7.6, 100 mM KOAc, 2 mM MgCl₂, 75 μg/ml BSA, 1 mM DTT) was mixed. The reaction was started by addition of ATP/Mg mix (1 mM final) and incubated for 120 min at 30°C, 300 rpm (Thermoshaker Comfort, Eppendorf). 1 μg of herring sperm DNA was added to chelate the remodeler for 5 min at 30°C before the samples were subjected to native gel electrophoresis. For the eviction assay, the reaction of the remodeling assay was supplemented with excess Nap1 (5 μM) to capture free-histone H2A–H2B dimer and H3–H4 tetramer.

## NanoDSF

For NanoDSF, protein samples were diluted to a concentration of 0.1–0.2 mg/ml and triplicate measurements in glass capillaries (Prometheus NT.48 Capillaries, PR-C002; Nanotemper Technologies) were performed on a Prometheus NT.48 (Nanotemper Technologies) over a temperature gradient from 20–90°C with a rate of +1°C/min. Results were evaluated with the PR.ThermControl software (v2.1.2).

## Cross-linking mass spectrometry

For crosslinking, 20 μg of Fun30 protein was crosslinked with 100x molar excess of BS3 for 30 min at 25°C before stopping the reaction by adding Tris pH 7.5 (final concentration 100 mM). For the mass spectrometry, crosslinked proteins were diluted 1:1 with digestion buffer (8 M Urea, 40 mM CAA, 10 mM TCEP, 50 mM Tris) and incubated for 20 min at 37°C followed by a 1:4 dilution with water. Crosslinked proteins were digested overnight at 37°C by addition of 0.5 μg of LysC and 1 μg of trypsin (Promega). The digestion was stopped by addition of 1% of TFA followed by desalting of the peptides using Sep-Pak C18 1cc vacuum cartridges (Waters). Desalted peptides were vacuum-dried.

Vacuum-dried peptides were dissolved at a concentration of 100 ng/μl in buffer A (0.1% formic acid). Peptides (100 ng) were separated and measured at a flow rate of 250 nl/min using the Thermo Easy-nLC 1200 (Thermo Fisher Scientific) coupled to the Orbitrap Exploris 480 mass spectrometer (Thermo Fisher Scientific). Peptides were separated on a 30-cm analytical column (inner diameter: 75 microns; packed in-house with ReproSil-Pur C18-AQ 1.9-micron beads, Dr. Maisch GmbH) using an increasing percentage of buffer B (80% acetonitrile, 0.1% formic acid). A linear gradient from 5–30% buffer B over 40 min, to 95% B over 10 min was used, and elution strength was held at 95% B for 5 min. The mass spectrometer was operated in the data-dependent mode with survey scans from m/z 300 to 1650 Th (resolution of 60 k at m/z = 200 Th). Up to 15 of the most abundant precursors were selected and fragmented using stepped (higher-energy C-trap dissociation with normalized collision energy of values of 19, 27, 35). The MS2 spectra were recorded with a dynamic m/z range (resolution of 30 k at m/z = 200 Th). AGC targets for MS1 and MS2 scans were set to 3 × 106 and 105, respectively, within a maximum injection time of 100 and 60 ms for the MS1 and MS2 scans. Charge state 2 was excluded from fragmentation.

The acquired raw data were processed using Proteome Discoverer (version 2.5.0.400; Thermo Fisher Scientific) with the XlinkX/PD nodes integrated. "NonCleavable" was set as acquisition strategy. The database search was performed against a FASTA containing the sequence(s) of the protein(s) under investigation and a contaminant database. DSS/BS3 was set as a crosslinker, cysteine carbamidomethylation was set as fixed modification and methionine oxidation and protein N-term acetylation were set as dynamic modifications. Trypsin/P was specified as protease and up to two missed cleavages were allowed. Identifications were accepted with a minimal score of 40 and a minimal δ score of 4. Filtering at 1% false discovery rate at the peptide level was applied by the XlinkX Validator node with setting simple.

Experimentally obtained crosslinks were visualized onto a 2D representation of the protein using xiNet (Combe et al, 2015) or onto the 3D model (AlphaFold2, Jumper et al, 2021) using PyMol v2.5.2 (The PyMOL Molecular Graphics System, Version 2.5.2, Schrödinger, LLC) with the plugin PyXlinkViewer (Schiffrin et al, 2020).

For the 3D mapping of the crosslinks onto the model, likely mobile regions without predicted secondary structure (±2 residues) were excluded. The threshold for BS3 crosslinks was set to 35 Å, allowing some flexibility taking into account that AlphaFold models may reflect an in-between situation of nucleotide-bound and apo-state of the enzyme.

## Multiple-sequence alignment of Fun30-SMARCAD1-ETL family

Multiple-sequence alignment of Fun30/SMARCAD1/ETL was performed using the software JalView v2.11.2.4. Full-protein sequences were aligned using ClustalWS.

## AlphaFold2 prediction and structural alignments with nucleosome remodelers

The respective protein sequences were submitted to AlphaFold2 (Jumper et al, 2021): Fun30 (1–1,131), Fun30ΔSAM (1–968), SMARCAD1 (1–1,026), Fun30 SAM-key (275–436), SMARCAD1 SAM-key region (203–488). The in trans complementation scenario was modelled using the AlphaFold2 multimer algorithm, providing the respective constructs Fun30ΔSAM and SAM-key as separate polypeptide

chains. Models were visualized using UCSF ChimeraX (Pettersen et al, 2021).

The AlphaFold2 model of Fun30 obtained from the AlphaFold Protein Structure Database (Varadi et al, 2022) was used for docking analysis. Extended regions with a low confidence score were rejected (residues 1–275, 410–560, 1,126–1,131). Structures were visualized and superimposed using UCSF ChimeraX (Pettersen et al, 2021).

For docking of the Fun30 model at the dyad, the nucleosomal DNA at SHL 2 of the RSC-bound nucleosome structure (PDB: 6TDA) was manually aligned with the dyad of a nucleosome core particle (PDB: 7OHC). Subsequently, the DNA of the RSC-bound nucleosome (PDB: 6TDA) and the nucleosome core particle (PDB: 7OHC) were fit into the cryo-EM map of the nucleosome core particle (EMD-12900), resulting in an improved alignment of SHL 2 of PDB: 6TDA with the dyad of PDB: 7OHC. The Fun30 AlphaFold model was aligned with the Sth1 ATPase (PDB: 6TDA) and the Fun30 model was visualized together with the nucleosome core particle (PDB: 7OHC).

For docking of the Fun30 model at SHL 2, the Fun30 AlphaFold model was superimposed with the Sth1 ATPase (PDB: 6TDA) by alignment of the ATPase N-lobes (Fun30 residues 561–802). The Fun30 model was visualized together with the nucleosome (PDB: 6TDA).

For comparison of the Fun30 model with Ino80 bound at SHL−6, a nucleosome-bound INO80 model was generated based on PDB: 8AV6 and EMD-15211 (Kunert et al, 2022). In brief, the structure of a nucleosome-bound INO80 complex (PDB: 8AV6) was fitted into the low-resolution cryo-EM map of INO80 bound to a nucleosome and extranuclesomal DNA (EMD-15211). The model was extended by fitting of extranucleosomal DNA and the post-HSA/HSA helix into the low-resolution cryo-EM map. For docking of the Fun30 model at SHL−6, the Fun30 AlphaFold model was superimposed with the Ino80 ATPase (model based on PDB: 8AV6 and EMD-15211) by alignment of the ATPase N-lobes (Fun30 residues 561–802). The Fun30 model was visualized together with the nucleosome (model based on PDB: 8AV6 and EMD-15211).

# Data Availability

XL-MS data have been deposited to the ProteomeXchange Consortium via the PRIDE partner repository with accession number PXD037249.

# Supplementary Information

# Acknowledgements

We thank members of the Pfander laboratory for discussions and/or critical reading of the study; S Mitzkus, I Kosmidou, P Heisig, and T Schmitt for technical support and providing preliminary data; Philipp Korber, Felix Müller-Planitz for protocols and experimental advice; Stefan Übel (MPIB biochemistry core facility) for nanoDSF and other biophysical measurements; Barbara Steigenberger (MPIB proteomics core facility) for XL-MS measurments and analysis; Daniel Bollschweiler (MPIB cryoEM facility) for mass photometry; Stefan Pettera (MPIB biochemistry core facility) for peptide synthesis. This work was supported by the Max Planck Society (to B Pfander), grants by the Deutsche Forschungsgemeinschaft (DFG, German Research Foundation): Project-ID 213249687—SFB 1064 (A23 to B Pfander, A06 to K-P Hopfner and A26 to CF Kurat) and by the European Research Council (INO3D: Number 833613 to K-P Hopfner).

## Author Contributions

LA Karl: conceptualization, resources, data curation, formal analysis, validation, investigation, visualization, methodology, and writing—original draft, review, and editing.
L Galanti: conceptualization, formal analysis, investigation, methodology, and writing—review and editing.
SCS Bantele: formal analysis, investigation, methodology, and writing—review and editing.
F Metzner: investigation, methodology, and writing—review and editing.
B Šafarić: resources.
L Rajappa: resources.
B Foster: resources and methodology.
VB Pires: investigation and methodology.
P Bansal: resources.
E Chacin: resources.
J Basquin: formal analysis and investigation.
KE Duderstadt: resources.
CF Kurat: resources.
T Bartke: resources, methodology, and writing—review and editing.
K-P Hopfner: resources, funding acquisition, investigation, methodology, and writing—review and editing.
B Pfander: conceptualization, resources, formal analysis, supervision, funding acquisition, investigation, visualization, methodology, project administration, and writing—original draft, review, and editing.

### Conflict of Interest Statement

The authors declare that they have no conflict of interest.

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
