## [Reviewer comments · Life Science Alliance]

Life Science Alliance

A SAM-key domain required for enzymatic activity of the Fun30 nucleosome remodeler

Leonhard Karl, Lorenzo Galanti, Susanne Bantele, Felix Metzner, Barbara Šafarić, Lional Rajappa, Benjamin Foster, Vanessa Pires, Priyanka Bansal, Erika Chacin, Jerome Basquin, Karl Duderstadt, Christoph Kurat, Till Bartke, Karl-Peter Hopfner, and Boris Pfander

DOI: <https://doi.org/10.26508/lsa.202201790>

Corresponding author(s): *Boris Pfander, German Aerospace Center*

Review Timeline:

Submission Date:	2022-10-31
Editorial Decision:	2022-12-08
Revision Received:	2023-06-12
Editorial Decision:	2023-06-16
Revision Received:	2023-06-28
Accepted:	2023-06-29

Transaction Report:

December 8, 2022

Re: Life Science Alliance manuscript #LSA-2022-01790-T

Dr. Boris Pfander
German Aerospace Center (DLR)
Institute of Aerospace Medicine, Genome Maintenance in Health and Disease
Linder Höhe
Cologne 51170
Germany

Dear Dr. Pfander,

Thank you for submitting your manuscript entitled "A SAM-key domain required for enzymatic activity and allosteric activation of the Fun30 nucleosome remodeler" to Life Science Alliance. The manuscript was assessed by expert reviewers, whose comments are appended to this letter. We invite you to submit a revised manuscript addressing the Reviewer comments.

Thank you for this interesting contribution to Life Science Alliance. We are looking forward to receiving your revised manuscript.

Sincerely,

B. MANUSCRIPT ORGANIZATION AND FORMATTING:

Reviewer #1 (Comments to the Authors (Required)):

Karl et al. investigate the role of a domain of the *S. cerevisiae* chromatin remodeler Fun30 in the context of chromatin remodeling and ATPase activity using genetic, biochemical, structural modeling and crosslinking mass spectrometry approaches. Fun30 belongs to the SMARCA4/Fun30 family of chromatin remodelers and plays roles in DNA damage response and gene silencing. It is a single-subunit chromatin remodeler and how its activity is regulated remains unclear. The authors identify an evolutionary conserved region in Fun30 that they term the SAM-key. SAM-key shares similarity to the post-HSA helix of Ino80 complexes. Deletion of the SAM-key region results in DNA damage response and gene silencing phenotypes in vivo. Biochemical characterization of the region demonstrates its importance for ATP hydrolysis and nucleosome sliding. Karl et al. show through structural modeling using AlphaFold2 and crosslinking mass spectrometry that the SAM-key region interacts intramolecularly with a region of Fun30 that is important for the activation of the remodeler. Overall, Karl et al. provide important information for our understanding of the intramolecular regulation of chromatin remodelers. Their findings are broadly applicable to multiple chromatin remodeling families. The presented data is of good quality and the authors make use of a broad range of assays to validate their findings. Only the gel shift assays for determining protein binding are limited in their interpretability and should be exchanged for fluorescence anisotropy measurements or similar solution-based techniques.

There is only one major concern that the authors must address:

1. The interpretability of the gel-based binding assays (e.g., presented in Fig. 2A and B and Fig. 6 and others) are strongly limited by the insufficiencies of the experimental setup. The gel shift assays do not seem to work well for Fun 30. This can lead to wrong interpretation of the results as no obvious and clearly defined shifted bands (corresponding to Fun30-containing DNA or nucleosome complex) can be observed. It is possible that aggregation of protein rather than actual binding of Fun30 constructs to the binding substrates leads to the observed binding effects. Because the authors are in the possession of labeled nucleosomes and DNA they should perform fluorescence anisotropy measurements or use a similar solution based assay.

Minor concerns:

2. The authors should provide a histogram of all crosslinks by distance. This makes it easier to judge how many of the crosslinks do not satisfy the length restraints of BS3.
3. The authors should better demonstrate where the closeup interactions in Fig. 4B map to the overall model.
4. The authors should show the full chromatogram of their purifications Fig. S2B. Why is there an additional much higher signal later in the trace?
5. Spelling of docket should be docked (Fig. 5)

Reviewer #2 (Comments to the Authors (Required)):

The paper by Karl et al gathers an impressive amount of work aimed at elucidating the mechanism of action of the *S. cerevisiae* chromatin remodeler Fun30, as a representative of the Fun30/SMARCA4/ETL class of remodeler. The relevance of studying Fun30 is that it acts on its own, while other remodelers include many subunits such as the INO80 complex. The authors cleverly addressed the function of Fun30 first by dissociating its DNA binding property from its remodeling function by directly addressing Fun30 to chromatin through a fusion with Ddc1 that they previously characterized. From there, they could identify a domain that they called the SAM-key domain that appears to be key for Fun30 chromatin remodeling activity. Notably, they could beautifully show that the SAM-key domain alone is able to mediate (when in large excess) the allosteric activation of full length Fun30 (be careful about proper referencing to figures, ie fig 3I and K for instance do not seem to be referred to properly).

Overall, with my molecular biology expertise only, I am impressed by the paper that looks great from the beginning to the end. This is the first time for me to have not a single concern about a paper.

Typos:

"the isolated SAK-key domain" => the isolated SAM-key domain
"INO80) stongly suggest the existence" => INO80) stongly suggests the existence

References:

"many fun30 phenotypes can be suppressed by mutation of the resection inhibitor Rad9 (Coestelloe et al 2012; Bentele et al 2017)": Chen et al 2012 showed a connection to Rad9 but not Costelloe et al 2012

Reviewer #3 (Comments to the Authors (Required)):

In the submitted manuscript, Karl et al. identify a SAM-key domain in the ATP-dependent chromatin remodeler Fun30/SMARCD1, an enzyme involved in DNA-damage repair and heterochromatin maintenance. They take advantage of AlphaFold2 for structure prediction and crosslinking mass-spec for structure verification. Their experiments confirm that, as the AF2 model shows, SAM-key is located between the two conserved ATPase lobes, implying its role in controlling the enzymatic activity. They subsequently generate a Fun30 construct lacking SAM-key, and show that the remodeler can bind to nucleosomes, but is deficient in the DNA-stimulated ATPase activity, sliding and eviction. Moreover, the authors also show that mutations introduced at the interface between SAM-key and ATPase lobes confirms the modulatory role of the former through allosteric mechanisms.

Had the authors taken the generative model in good faith, it could have been seen as overly optimistic; However, the cautiousness exerted by the authors towards the model, and follow-up XL-MS experiments as a confirmation of interactions is a correct approach.

Authors subsequently generate: a synthetic construct lacking the SAM-key domain, and constructs with mutations at the SAM-key/ATPase-lobe interface. The experiments with the deltaSAM show a clear attenuation of ATPase activity and sliding, confirming the ML/XL-MS model. The authors point to the similarity of the SAM-key domain to HSA/post-HSA domains of Ino80 and Sth1. They further conclude that the interaction between protrusion I, a subdomain located between the two ATPase lobes, and SAM-key, is not involved in nucleosome binding, but that the SAM-key and its interaction with protrusion I is required for SMARCD1/Fun30 catalytic activity.

The authors correctly conclude that the allosteric modulation of the remodeler activity could be the distinguishing element in targeting this remodeler with more specific small molecules, as the ATPase lobes are generally conserved across remodelers.

I see no issue with the manuscript and recommend it for publication.

Point-by-point response: Karl et al.

Reviewer 1

Karl et al. investigate the role of a domain of the *S. cerevisiae* chromatin remodeler Fun30 in the context of chromatin remodeling and ATPase activity using genetic, biochemical, structural modeling and crosslinking mass spectrometry approaches. Fun30 belongs to the SMARCAD1/Fun30 family of chromatin remodelers and plays roles in DNA damage response and gene silencing. It is a single-subunit chromatin remodeler and how its activity is regulated remains unclear. The authors identify an evolutionary conserved region in Fun30 that they term the SAM-key. SAM-key shares similarity to the post-HSA helix of Ino80 complexes. Deletion of the SAM-key region results in DNA damage response and gene silencing phenotypes in vivo. Biochemical characterization of the region demonstrates its importance for ATP hydrolysis and nucleosome sliding. Karl et al. show through structural modeling using AlphaFold2 and crosslinking mass spectrometry that the SAM-key region interacts intramolecularly with a region of Fun30 that is important for the activation of the remodeler.

Overall, Karl et al. provide important information for our understanding of the intramolecular regulation of chromatin remodelers. Their findings are broadly applicable to multiple chromatin remodeling families. The presented data is of good quality and the authors make use of a broad range of assays to validate their findings. Only the gel shift assays for determining protein binding are limited in their interpretability and should be exchanged for fluorescence anisotropy measurements or similar solution-based techniques.

Response:

We thank the reviewer for the insightful comments and constructive criticism, which we have addressed in their entirety and using new experiments. We are therefore confident that the reviewer will find our paper much improved and suitable for publication.

There is only one major concern that the authors must address:

1. The interpretability of the gel-based binding assays (e.g., presented in Fig. 2A and B and Fig. 6 and others) are strongly limited by the insufficiencies of the experimental setup. The gel shift assays do not seem to work well for Fun 30. This can lead to wrong interpretation of the results as no obvious and clearly defined shifted bands (corresponding to Fun30-containing DNA or nucleosome complex) can be observed. It is possible that aggregation of protein rather than actual binding of Fun30 constructs to the binding substrates leads to the observed binding effects. Because the authors are in the possession of labeled nucleosomes and DNA they should perform fluorescence anisotropy measurements or use a similar solution based assay.

Response:

We followed the reviewer's suggestion and performed the suggested fluorescence anisotropy measurement (Reviewer Figure 1). We tested different buffer conditions as well as various DNA substrates. We observe slight changes in the anisotropy signal, which could indicate DNA binding by Fun30. However, these changes are too small to confidently interpret them as Fun30-DNA binding.

To address the main concern of the reviewer that the observed gel shift may be due to Fun30-DNA aggregation, we tested reversibility of the gel shift. Specifically, we added excess amounts of a competitor DNA (herring sperm DNA, Fig. S2F), which reversed Fun30 binding to the DNA fragment. Therefore, we conclude that we are observing reversible DNA binding and not irreversible DNA-protein aggregation.

Minor concerns:

2. The authors should provide a histogram of all crosslinks by distance. This makes it easier to judge how many of the crosslinks do not satisfy the length restraints of BS3.

Response:

The histograms are now included in Supplementary Figure 4D.

3. The authors should better demonstrate where the closeup interactions in Fig. 4B map to the overall model.

Response:

We updated figures 4 and S4 and included a frame to indicate the location of the close-up.

4. The authors should show the full chromatogram of their purifications Fig. S2B. Why is there an additional much higher signal later in the trace?

Response:

We now included the full chromatogram in the Figure S2B. The additional higher signal later in the trace is due to the cleaved-off 6xHis-GST-affinity tag and the His-tagged 3C-protease which were removed by size exclusion chromatography.

5. Spelling of docket should be docked (Fig. 5)

Response:

We corrected the spelling mistake.

Reviewer 2

The paper by Karl et al gathers an impressive amount of work aimed at elucidating the mechanism of action of the *S. cerevisiae* chromatin remodeler Fun30, as a representative of the Fun30/SMARCD1/ETL class of remodeler. The relevance of studying Fun30 is that it acts on its own, while other remodelers include many subunits such as the INO80 complex. The authors cleverly addressed the function of Fun30 first by dissociating its DNA binding property from its remodeling function by directly addressing Fun30 to chromatin through a fusion with Ddc1 that they previously characterized. From there, they could identify a domain that they called the SAM-key domain that appears to be key for Fun30 chromatin remodeling activity. Notably, they could beautifully show that the SAM-key domain alone is able to mediate (when in large excess) the allosteric activation of full length Fun30 (be careful about proper referencing to figures, ie fig 3I and K for instance do not seem to be referred to properly).

Overall, with my molecular biology expertise only, I am impressed by the paper that looks great from the beginning to the end. This is the first time for me to have not a single concern about a paper.

Response:

We are delighted to hear and thank the reviewer for praising our manuscript in this form.

Typos:

"the isolated SAK-key domain" => the isolated SAM-key domain

"INO80) stongly suggest the existence" => INO80) stongly suggests the existence

Response: Corrected.

References:

"many fun30 phenotypes can be suppressed by mutation of the resection inhibitor Rad9 (Coestelloe et al 2012; Bentele et al 2017)": Chen et al 2012 showed a connection to Rad9 but not Costelloe et al 2012

Response: Corrected.

Reviewer 3:

In the submitted manuscript, Karl et al. identify a SAM-key domain in the ATP-dependent chromatin remodeler Fun30/SMARCAD1, an enzyme involved in DNA-damage repair and heterochromatin maintenance. They take advantage of AlphaFold2 for structure prediction and crosslinking mass-spec for structure verification. Their experiments confirm that, as the AF2 model shows, SAM-key is located between the two conserved ATPase lobes, implying its role in controlling the enzymatic activity. They subsequently generate a Fun30 construct lacking SAM-key, and show that the remodeler can bind to nucleosomes, but is deficient in the DNA-stimulated ATPase activity, sliding and eviction. Moreover, the authors also show that mutations introduced at the interface between SAM-key and ATPase lobes confirms the modulatory role of the former through allosteric mechanisms.

Had the authors taken the generative model in good faith, it could have been seen as overly optimistic; However, the cautiousness exerted by the authors towards the model, and follow-up XL-MS experiments as a confirmation of interactions is a correct approach.

Authors subsequently generate: a synthetic construct lacking the SAM-key domain, and constructs with mutations at the SAM-key/ATPase-lobe interface. The experiments with the deltaSAM show a clear attenuation of ATPase activity and sliding, confirming the ML/XL-MS model. The authors point to the similarity of the SAM-key domain to HSA/post-HSA domains of Ino80 and Sth1. They further conclude that the interaction between protrusion I, a subdomain located between the two ATPase lobes, and SAM-key, is not involved in nucleosome binding, but that the SAM-key and its interaction with protrusion I is required for SMARCAD1/Fun30 catalytic activity.

The authors correctly conclude that the allosteric modulation of the remodeler activity could be the distinguishing element in targeting this remodeler with more specific small molecules, as the ATPase lobes are generally conserved across remodelers.

I see no issue with the manuscript and recommend it for publication.

Response: We are delighted to hear and thank the reviewer for the highly positive feedback.

June 16, 2023

RE: Life Science Alliance Manuscript #LSA-2022-01790-TR

Dr. Boris Pfander
German Aerospace Center
Institute of Aerospace Medicine, Genome Maintenance in Health and Disease
Linder Höhe
Cologne 51170
Germany

Dear Dr. Pfander,

Thank you for submitting your revised manuscript entitled "A SAM-key domain required for enzymatic activity of the Fun30 nucleosome remodeler". We would be happy to publish your paper in Life Science Alliance pending final revisions necessary to meet our formatting guidelines.

- please upload all figure files as individual ones, including the supplementary figure files; all figure legends should only appear in the main manuscript file
- please upload your Supplementary Tables in editable .doc or excel format;
- please add the Twitter handle of your host institute/organization as well as your own or/and one of the authors in our system
- the titles on the system and manuscript file should match
- please make sure the author order in your manuscript and our system match; the full name (middle names as initials) of each author should be given on the title page
- please upload the manuscript file without tracking changes
- please add your main, supplementary figure, and table legends to the main manuscript text after the references section
- the Supplemental References should be incorporated into the main Reference list
- please use the [10 author names et al.] format in your references (i.e., limit the author names to the first 10)
- please add a callout for Table S3 to your main manuscript text

A. FINAL FILES:

B. MANUSCRIPT ORGANIZATION AND FORMATTING:

Sincerely,

June 29, 2023

RE: Life Science Alliance Manuscript #LSA-2022-01790-TRR

Dr. Boris Pfander
German Aerospace Center
Institute of Aerospace Medicine, Genome Maintenance in Health and Disease
Linder Höhe
Cologne 51170
Germany

Dear Dr. Pfander,

Thank you for submitting your Research Article entitled "A SAM-key domain required for enzymatic activity of the Fun30 nucleosome remodeler". It is a pleasure to let you know that your manuscript is now accepted for publication in Life Science Alliance. Congratulations on this interesting work.

DISTRIBUTION OF MATERIALS:

Again, congratulations on a very nice paper. I hope you found the review process to be constructive and are pleased with how the manuscript was handled editorially. We look forward to future exciting submissions from your lab.

Sincerely,
